# VEGF signaling regulates the fate of obstructed capillaries in mouse cortex

**Patrick Reeson[1], Kevin Choi[1], Craig E Brown[1,2,3]\***

[1]Division of Medical Sciences, University of Victoria, Victoria, Canada; [2]Department of Biology, University of Victoria, Victoria, Canada; [3]Department of Psychiatry, University of British Columbia, Vancouver, Canada

**Abstract** Cortical capillaries are prone to obstruction, which over time, could have a major impact on brain angioarchitecture and function. The mechanisms that govern the removal of these obstructions and what long-term fate awaits obstructed capillaries, remains a mystery. We estimate that ~0.12% of mouse cortical capillaries are obstructed each day (lasting >20 min), preferentially in superficial layers and lower order branches. Tracking natural or microsphere-induced obstructions revealed that 75–80% of capillaries recanalized within 24 hr. Remarkably, 30% of all obstructed capillaries were pruned by 21 days, including some that had regained flow. Pruning involved regression of endothelial cells, which was not compensated for by sprouting. Using this information, we predicted capillary loss with aging that closely matched experimental estimates. Genetic knockdown or inhibition of VEGF-R2 signaling was a critical factor in promoting capillary recanalization and minimizing subsequent pruning. Our studies reveal the incidence, mechanism and long-term outcome of capillary obstructions which can also explain age-related capillary rarefaction.
DOI: https://doi.org/10.7554/eLife.33670.001

## Introduction

The brain is an energetically demanding organ that contains kilometers of capillaries to meet this need. Capillaries are the smallest vessels in the brain and serve as the primary site of nutrient and gas exchange (*Attwell et al., 2010*). These capillary networks are critical for maintaining proper brain function since decrements in cognition that occur with aging and certain neurodegenerative diseases (like Alzheimer's disease) are associated with the loss of brain capillaries (*Brown and Thore, 2011*; *Iadecola, 2013*; *Riddle et al., 2003*). From the earliest in vivo imaging studies of cerebral blood flow over two decades ago, the susceptibility of capillaries to obstruction, even in healthy animals has been noted (*Kleinfeld et al., 1998*; *Santisakultarm et al., 2014*; *Villringer et al., 1994*). A recent study used Optical Coherence Tomography to estimate that over 9 min, up to 7.5% of capillaries experienced a stall (*Erdener et al., 2017*). This is not surprising since capillaries are inherently narrow (~3–5 µm diameter) high resistance tubes that must pass relatively large and adherent components in the blood (red blood cells, leukocytes, cholesterol, fibrin etc.). Cortical capillaries also experience the largest drop in pressure across the cerebral vasculature (*Gould et al., 2017*). However, since the initial in vivo observations, no study has comprehensively followed the long-term outcome of capillary obstructions and what mechanisms dictate their fate. This leaves open many questions that are critical to our understanding of how mature microvascular networks change. For example, if obstructions are almost always cleared or compensated for with collateral sprouting, then the impact of these obstructions could be minimal. If, however, obstructions lead to the pruning of vessel segments, then it is conceivable that the accumulation of these events over a time could drive the progressive rarefaction of cerebrovascular networks commonly found with aging (*Mann et al., 1986*; *Klein and Michel, 1977*; *Hinds and McNelly, 1982*; *Casey and Feldman, 1985*;

**\*For correspondence:** brownc@uvic.ca

*Buchweitz-Milton and Weiss, 1987*; *Jucker et al., 1990*; *Amenta et al., 1995a*) and certain neuro-degenerative diseases (*Brown and Thore, 2011*; *Iadecola, 2013*; *Tong and Hamel, 2015*).

In order to understand what ultimately becomes of obstructed cortical capillaries and the mechanisms that regulate their fate, we employed in vivo time-lapse imaging to identify and follow spontaneous, naturally occurring obstructions in mouse cortex. Due to the unpredictable and sparse nature of these obstructions, we also developed and validated an experimental model of inducing capillary obstructions with fluorescent microspheres. Although we find that the majority of capillary obstructions are cleared, almost one-third of obstructed capillaries are pruned without compensatory angiogenesis. Based on this information and the relative incidence of naturally occurring obstructions, we were able to model capillary loss with aging. Furthermore, we show that VEGF-R2 (encoded by *Kdr*) signaling, a known sensor of shear stress, is critical for mediating clearance of obstructions and preventing capillary pruning. Our findings shed new light on the long-term fates of clogged cortical capillaries and the mechanisms that dictate this process.

## Results

### Superficial and lower order cortical capillaries are prone to obstruction

We first examined how frequent and what parts of the vascular tree were prone to spontaneous obstructions in the mouse somatosensory cortex. We imaged cortical vascular networks (~300 μm in depth) in both lightly anesthetized (2% isoflurane for induction, 1% for maintenance) and awake *Tek*-GFP mice injected with Rhodamine B to visualize blood flow. We could identify flowing and non-flowing capillaries by the presence or absence of streaking red blood cells (RBCs) (*Figure 1A*). Spontaneous naturally occurring obstructions typically presented with a gap in capillary lumen fluorescence (mean lumen diameter of 3.85 ± 1.25 μm) created by stalled cells or debris (*Santisakultarm et al., 2014*) (*Figure 1A*, insert, red arrow). Our analysis was restricted to capillary segments lacking flow for longer than 20 min to exclude transient obstructions that affect ~1–7% of capillaries as previously reported (*Santisakultarm et al., 2014*; *Erdener et al., 2017*). Unbiased sampling of cortical vasculature in 16 mice for 2 hr revealed that longer lasting obstructions (>20 min) were relatively rare, affecting 2 in 20,334 capillaries (~0.118% capillaries obstructed per day, *Figure 1—figure supplement 1*)(*Reeson, 2018*). While these spontaneous obstructions were rare, 10 of the 16 mice had non-flowing capillaries segments present at the start of imaging, indicating spontaneous obstructions were not unique to a few mice (persisted >20 min, 20 obstructed capillaries in 10 animals, range of 1–4 per mouse). To confirm this, we injected mice with 1 μm microspheres coated in lipophilic dye DiI solution. Microspheres of this size freely circulate in the blood but become stalled in spontaneously obstructed capillaries, allowing the DiI to leech into the endothelium, creating an indelible stamp of the spontaneous obstruction. Quantifying postmortem DiI-labeled capillaries 3 hr after injection, we found 3.69 ± 0.97 obstructed capillaries per $mm^3$ of cortex. Based on estimates of ~20,000 capillaries per $mm^3$ (*Blinder et al., 2013*; *Tsai et al., 2009*) this gives a rate of 0.14% spontaneous obstructions per day, nearly identical to our in vivo estimate. Furthermore, the rate of obstructions did not differ between awake and isoflurane anesthetized mice [% volume obstructed $t_{(14)}$=1.09, p=0.29, % length obstructed $t_{(14)}$=1.79, p=0.10, $n_{(isoflurane)}$=11, $n_{(awake)}$=5 mice] thereby ruling out spontaneous obstructions as purely artifacts of anesthetics.

Due to the rare and spontaneous nature of capillary obstructions, we experimentally modeled these events by injecting 4 μm diameter fluorescent microspheres (i.v.; *Figure 1 – figure supplement 2A, B* see Materials and methods). Although microspheres accumulated in peripheral organs, particularly the liver (*Figure 1—figure supplement 2C*), injections did not induce any significant changes in body weight, cardiovascular function, hematological chemistry or hematocrit (*Figure 1—figure supplement 2D–J*), nor did it lead to inflammatory microglia responses or cell death in either the cortex or peripheral organs (*Figure 1—figure supplement 3*). In the brain, injection of microspheres was sufficient to obstruct 3.67% (95% CI 2.5–4.8%) of cortical capillaries (~30 min after injection). Although obstructions were spread across the cortex (in an anterior to posterior manner) and thus affected all major cerebral vascular territories (*Figure 1—figure supplement 4*), they occurred preferentially in more superficial layers (*Figure 1B*) and at lower branch order capillaries (*Figure 1C*). These observations were noted in both spontaneously occurring and microsphere-induced capillary

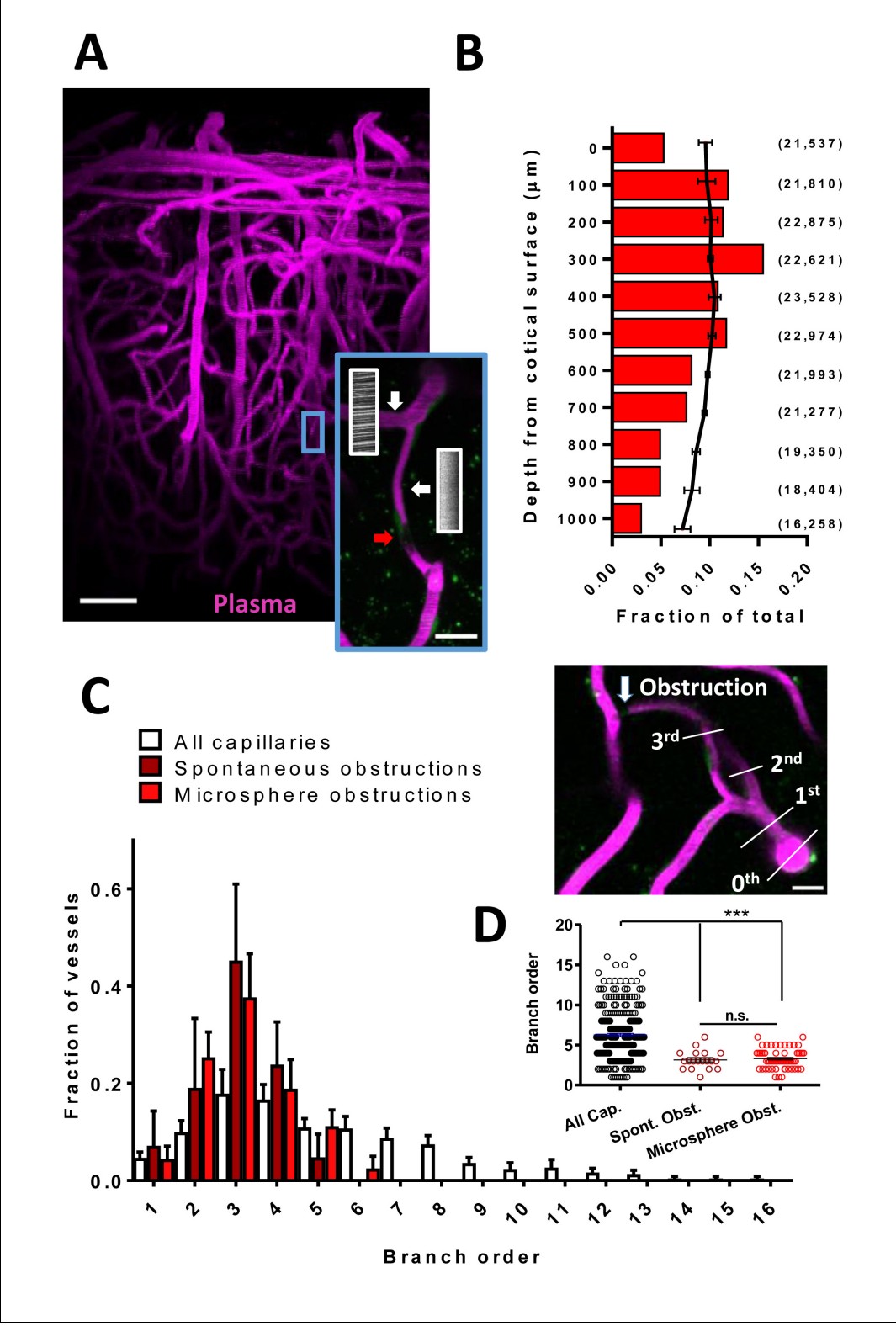

**Figure 1.** Cortical capillaries are prone to spontaneous obstruction. (**A**) Side projection from in vivo two-photon imaging stack (0.027 mm³) showing pial surface at the top with labeled plasma (magenta). Insert shows a flowing (Note streaking pattern in line scans caused by RBC movement) and obstructed capillary (no streaks). Red arrow indicates occluding debris/cells. Scale bars are 50 µm and 15 µm for inset. (**B**) Distribution of microsphere obstructed capillaries as it relates to depth from the pial surface determined by confocal imaging from post-

*Figure 1 continued on next page*

*Figure 1 continued*

mortem brain sections. Red bars indicate the relative amount of microsphere obstructions for each depth expressed as a fraction of total obstructions. Black line indicates fraction of total capillaries by depth (error bars are 95% CI) as well as raw numbers of capillaries/ mm$^3$ by depth are provided in parentheses. Note 0.04 obstructions occurred below 1000 μm from cortical surface but are not shown. (C) Distribution showing spontaneous and microsphere obstructed capillaries expressed as a function of arteriole branch order and relative to the distribution of all capillaries [n$_{(all)}$=3 mice, 285 capillaries; n$_{(spont.)}$=5 mice, 21 obstructions; n$_{(micro.)}$=5 mice, 59 obstructions). Note that lower order capillaries are more susceptible to obstruction. Inset illustrates branch order which started with the penetrating arteriole (0 order branch). Scale bar 15 μm. (D) Mean branch order of spontaneous or microsphere-induced capillary obstructions ([one way ANOVA F$_{(2,360)}$=32.36, p<0.0001, all capillaries compared to spontaneous obstructions unpaired t-test t$_{(302)}$=4.180, p<0.0001, or microsphere obstructions unpaired t-test t$_{(342)}$=6.95, p<0.0001; spontaneous vs microsphere obstructions t$_{(76)}$=0.46, p=0.65]. ***p<0.001, n.s. = not significant. Error bars are S.E.M.

DOI: https://doi.org/10.7554/eLife.33670.002

The following figure supplements are available for figure 1:

**Figure supplement 1.** Work flow and validation of automated estimates of vessel density.

DOI: https://doi.org/10.7554/eLife.33670.003

**Figure supplement 2.** Fluorescent microspheres as a model of spontaneous naturally occurring capillary obstructions.

DOI: https://doi.org/10.7554/eLife.33670.004

**Figure supplement 3.** Microsphere based obstruction and pruning did not induce a microglial response or cell death.

DOI: https://doi.org/10.7554/eLife.33670.005

**Figure supplement 4.** Microsphere obstructions are distributed across major cerebral vascular territories.

DOI: https://doi.org/10.7554/eLife.33670.006

obstructions (*Figure 1C,D*). Therefore, the risk of obstruction is significantly higher in a subset of superficial and lower arteriole branch order cerebral capillaries.

## Fates of obstructed cortical capillaries

We next assessed capillary recanalization rates and their fates using in vivo time lapse imaging. Imaging stacks were focused on regions where we could find natural/spontaneous or microsphere-induced obstructions to boost sampling rates. Over the 21-day imaging period, we found different outcomes for obstructed capillaries. First, all capillaries that did not recanalize were pruned away in a step wise fashion (*Franco et al., 2015*). Pruning progressed with a pinched endothelial segment at one end followed by the retraction of that segment (*Figure 2A*), leaving the remaining segment connected to the adjacent, flowing capillary. Retraction of vessel segments was associated with an increase in endothelial cell nuclei [unpaired t-test t$_{(29)}$=6.625, p<0.0001] around pruned branch points (*Figure 2B*), suggesting that endothelial cells regress and integrate into adjacent capillaries, reminiscent of endothelial regression found in development (*Franco et al., 2015*; *Chen et al., 2012*). During pruning we occasionally found that the adjacent capillary segments appeared closer together or further apart. However, by measuring inter-capillary distance at time 0 and +21 days, on average we found no significant lasting deformation of the newly separated capillaries compared to controls (*Figure 2—figure supplement 1*). To our surprise, pruning of vessel segments was never associated with sprouting of new capillaries (0 sprouts in 162 capillaries followed over 21 days). Therefore, compensatory sprouting of capillaries does not occur, or rarely to compensate for pruning.

For capillaries obstructed with microspheres, almost all recanalized by extruding the obstruction back into the circulation (*Figure 2C*, *Figure 2—figure supplement 2*, and *Figure 2—video 1*), while only a small fraction (2%) recanalized by extruding the microsphere through the endothelial wall into the parenchyma, commonly referred to as angiophagy (*Figure 2E*; *Figure 2—figure supplement 2* and *Figure 2—video 2*). However, we should note that we were unable to measure rates of angiophagy associated with spontaneous, naturally occurring emboli, which likely would have influenced the route of recanalization (washout vs angiophagy). Within the first 24 hr, 75–80% of capillaries (stalled >20 min at time 0) recanalized, which did not differ significantly between spontaneous and microsphere-induced obstructions (*Figure 2D*; effect of time: F$_{(4,40)}$=55.29, p<0.001, effect of

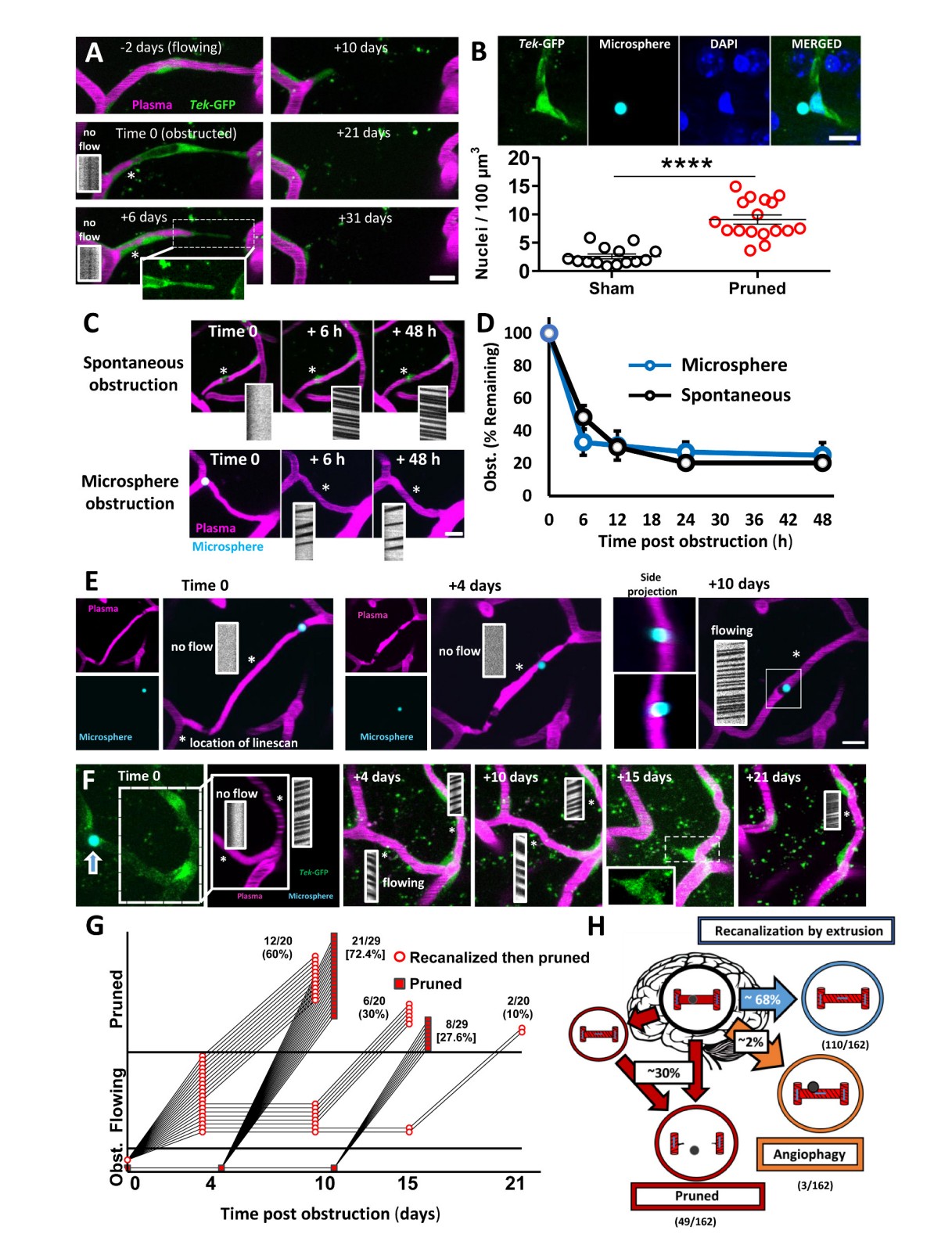

**Figure 2.** Fates of obstructed cortical capillaries. A) Longitudinal imaging of a spontaneously obstructed capillary (time 0) that failed to recanalize and was subsequently pruned over 31 days. The endothelium is shown in green (*Tek*-GFP) and blood plasma in magenta. Asterisk shows location of linescan. (B) Top: Confocal images show endothelial cell nuclei at a branching point adjacent to a presumptively obstructed and pruned capillary segment. Bottom: Pruning of capillaries was associated with a local (within 10 μm) increase in endothelial cell nucleus density [n = 4 mice, unpaired

*Figure 2 continued on next page*

*Figure 2 continued*

$t_{(29)}$=6.625, ****p<0.0001]. Average 4.6 capillaries per mouse, range 3–6. (**C**) Time lapse images of spontaneous or microsphere-induced capillary obstructions, both of which were cleared within 48 hr. D) Clearance of spontaneous (black) or microsphere-induced (blue) capillary obstructions over 48 hr as percent of obstructions at time 0. Recanalization rates were not significantly different between the two types of obstruction [two-way ANOVA, main effect of group: $F_{(1,40)}$=0.06, p=0.99; $n_{(spont.)}$=5 mice, 36 obstructions, $n_{(micro.)}$=4 mice, 35 obstructions]. (**E**) Example of a capillary that recanalized by extruding the obstruction through the vessel wall, also known as angiophagy. Note the return of blood flow at +10 days and the displacement of the microsphere in side image projections (also see 3D reconstruction, *Figure 2—video 2*). (**F**) Time lapse images reveal that some capillaries regain blood flow only to be eliminated at a later time. G) Time course of all capillaries that failed to recanalize and were pruned (square) compared to those that recanalized and were later pruned (circle). **H**) Summary of microsphere obstructed capillary fates 21 days after injection (n = 14 mice, 162 obstructions). Numbers in parentheses indicate how many capillaries out of 162 total capillaries underwent pruning, angiophagy or recanalized by extrusion. Average 11 capillaries per mouse, range 6–26. Scale bars = 10 μm. Error bars are S.E.M.

DOI: https://doi.org/10.7554/eLife.33670.007

The following video and figure supplements are available for figure 2:

**Figure supplement 1.** Capillary pruning does not alter adjoining capillary position.

DOI: https://doi.org/10.7554/eLife.33670.008

**Figure supplement 2.** Additional examples of capillary recanalization and pruning.

DOI: https://doi.org/10.7554/eLife.33670.009

**Figure supplement 3.** Recanalization correlates with obstruction location but not Local blood flow.

DOI: https://doi.org/10.7554/eLife.33670.010

**Figure supplement 4.** Blood flow in recanalized capillaries does not predict later pruning.

DOI: https://doi.org/10.7554/eLife.33670.011

**Figure supplement 5.** Obstructed capillaries that recanalized had a higher risk for subsequent obstruction.

DOI: https://doi.org/10.7554/eLife.33670.012

**Figure 2—video 1.** Capillary recanalization through extruding microsphere back into the circulation.

DOI: https://doi.org/10.7554/eLife.33670.013

**Figure 2—video 2.** Capillary recanalization through angiophagy.

DOI: https://doi.org/10.7554/eLife.33670.014

obstruction type: $F_{(4,40)}$=0.11, p=0.75). The only vascular factor that predicted recanalization was proximity of the obstruction to the nearest flowing capillary (*Figure 2—figure supplement 3A,B*). By contrast, capillary branch order, blood flow velocity or lumen diameter in upstream and downstream capillaries was not predictive of recanalization (*Figure 2—figure supplement 3C–G*). Interestingly by 21 days, 30% of all obstructed capillaries (at time 0) had been eliminated. While one would have expected 20–25% based on recanalization rates in the first 48 hr, we discovered that a subset of capillaries regained blood flow only to be pruned at a later time point (*Figure 2F,G*, *Figure 2—figure supplement 1A* and *Figure 2—figure supplement 2B*). This phenomenon could not be predicted by changes in blood flow velocity, width, or flux in the recanalized primary capillary, or connected (secondary) capillaries (*Figure 2—figure supplement 4*). However, by inducing a second wave of microsphere obstructions (microsphere injections 4 days apart), we noticed that the probability of experiencing an obstruction was greater in capillaries that were previously obstructed and regained flow (*Figure 2—figure supplement 5*). Therefore, it is possible that delayed pruning of capillaries in ones that initially regained blood flow, could be explained by a second spontaneous obstruction that was missed in the days between imaging (*Erdener et al., 2017*). In summary (*Figure 2H*), by following capillaries for 21 days after an obstruction, we determined there was a 69.8% chance the capillary would be intact and flowing either by extruding the emboli back into circulation (110/162 capillaries) or through angiophagy (3/162 capillaries). There was a 30.2% chance of being pruned (49/162 capillaries), with no evidence of compensatory sprouting. Importantly, the likelihood of capillary pruning did not differ significantly between obstructions that occurred spontaneously versus those that were induced with microspheres (unpaired t-test $t_{(19)}$=0.72, p=0.48).

## Impact of capillary pruning on local blood flow

From a hemodynamic perspective, capillary pruning led to a transient increase in blood flow velocity and flux (flux estimates volume of blood flowing per second through a vessel) in adjacent capillaries, particularly those upstream of the pruned segment (*Figure 3A–E*). By contrast, RBC velocity, vessel width and flux in control capillaries such as those from vehicle injected mice (without microspheres), capillaries distant to obstructions or those adjacent to recanalized capillaries were stable across

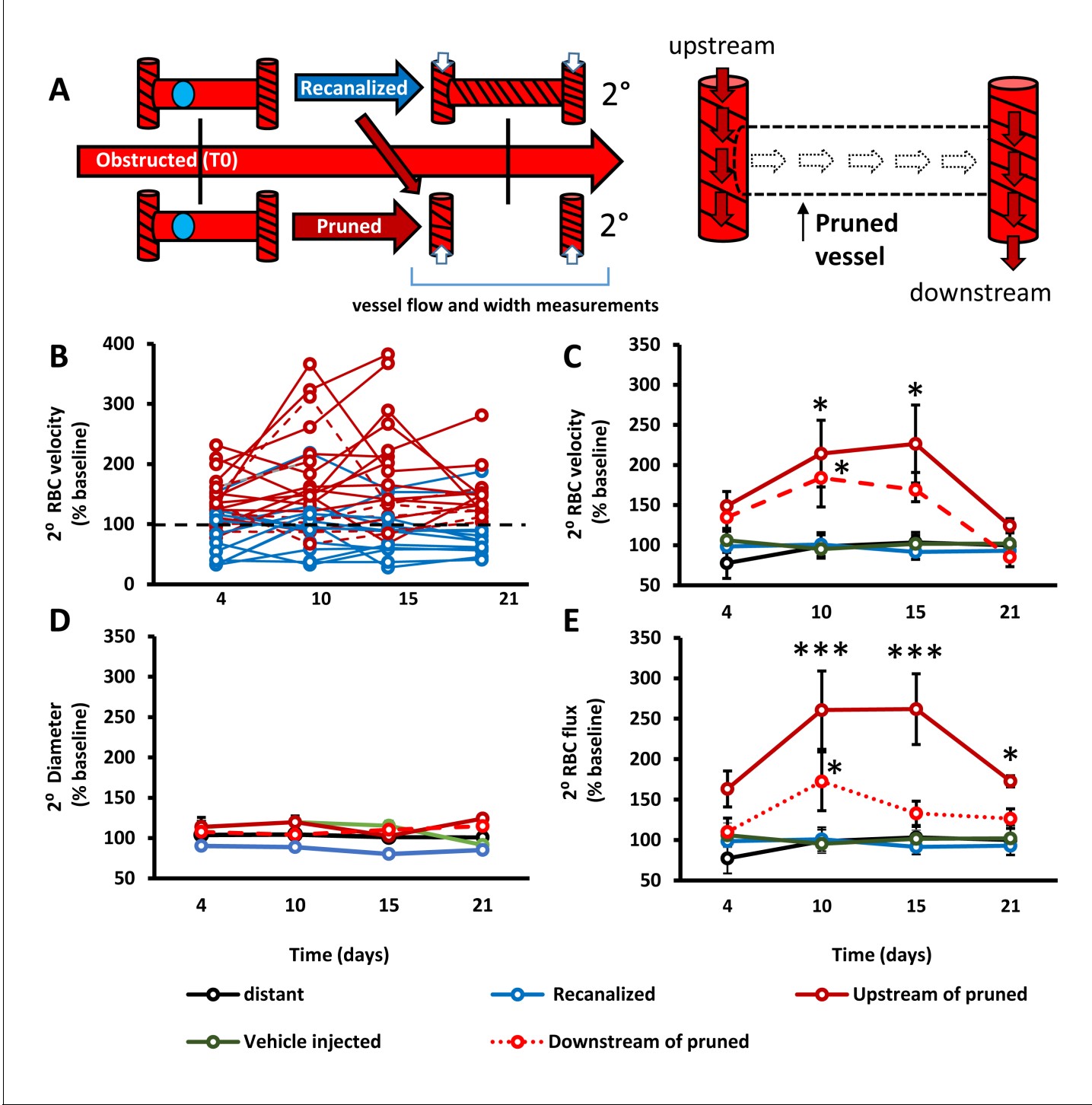

**Figure 3.** Capillary pruning leads to altered blood flow in adjacent connected capillaries. A) Diagram summarizing measurements taken from secondary/adjacent (2°) capillaries. (B) Plots show normalized RBC velocity from individual secondary capillaries after the primary segment recanalized (blue) or was pruned (solid or dashed red). C– -E) Graphs show mean normalized secondary RBC velocity (C); Main effect of Group $F_{(4,151)}=10.27$, p<0.0001), diameter (D); Main effect of group $F_{(4,118)}=2.31$, p=0.08) and RBC flux (E); Main effect of group $F_{(4,68)}=3.53$, p=0.03) in the 5 groups of capillaries studied. Note that 'vehicle injected' refers to capillaries tracked from mice injected with just vehicle solution and no microspheres (green) and 'distant vessels' refers to capillaries that were distant to any microsphere obstructed capillaries (black) (defined as minimum of 2 branching points away from obstruction). n = 14 mice, 6–20 vessels per group. Average 11 capillaries per mouse, range 4–20. Individual time points followed up with unpaired t-test. *p<0.05 compared to vehicle injected, **p<0.01, ***p<0.001. Error bars are S.E.M.
DOI: https://doi.org/10.7554/eLife.33670.015

imaging timepoints (*Figure 3B–E*). These results show that capillary pruning induces a relatively long-lasting (over several days) perturbation of blood flow in connected capillary networks.

## Lower capillary density in aged mice is predicted by obstruction and pruning rates

Next, we attempted to predict age-related changes in cortical capillary density based on our experimental estimates of spontaneous capillary obstructions (~0.118% capillaries obstructed per day) and the likelihood of pruning (30% of all obstructed capillaries). We first replicated well-established reports of an age-related reduction in capillary density (*Brown and Thore, 2011*; *Riddle et al., 2003*; *Klein and Michel, 1977*; *Hinds and McNelly, 1982*; *Casey and Feldman, 1985*; *Buchweitz-Milton and Weiss, 1987*; *Jucker et al., 1990*; *Amenta et al., 1995a*; *Jucker and Meier-Ruge, 1989*; *Amenta et al., 1995b*) by comparing vascular networks in young adult and aged *Tek*-GFP mice (*Figure 4A,B*; 3–4 vs. 15–18 month-old mice, respectively; 6 and 7 mice per group,>49,000 capillaries). Based on our analysis, we found a 10 ± 8% loss in capillary density over the 13.5 month age difference (*Figure 4C*). For theoretical predictions, we created a hypothetical population of 100,000 capillaries, each assigned a branch order based on our experimentally determined distribution (*Reeson, 2018*). We then assigned each branch order a risk of obstruction, based again on our experimental distribution (*Figure 1C*). For every 2 hr window the number of capillaries lost for each branch order was simply the number of capillaries x risk of obstruction for that branch order x risk of pruning (30%) (*Figure 4—figure supplement 1*). Comparing the fraction of vessel loss predicted by our model over the course of ~13.5 months (~405 days, with day 0 representing young adult mouse in *Figure 4C*), we found close agreement between experimentally observed and predicted capillary loss (predicted = 88% capillaries remaining vs. measured = 90 ± 8%; Unpaired t-test to hypothetical mean 0.88, $t_{(6)}$=0.68, p=0.52). The predicted loss matched some, but not all, published estimates of age related capillary loss in rats (*Figure 4—figure supplement 1*) Thus, spontaneous capillary obstruction and subsequent pruning without compensatory sprouting, can account for age related loss in capillary density.

## VEGF-R2 signaling dictates capillary recanalization

Since the maintenance of capillary network density is important for brain health (*Iadecola, 2013*), our next goal was to uncover a molecular mechanism dictating capillary recanalization. We focused on Vascular Endothelial Growth Factor Receptor 2 (VEGF-R2) signaling given that it is a critical regulator of endothelial cell function and is very sensitive to changes in hemodynamic shear stress (*Olsson et al., 2006*; *Tzima et al., 2005*). To assess relative levels of VEGF-R2 signaling within individual capillaries that were obstructed vs. those that recanalized, we injected 4 µm microspheres coated in a DiI solution and examined brains 3 hr after injection. As mentioned previously, lipophilic DiI leaches into endothelial cells at the site of the obstruction, therefore leaving an indelible stamp of where the obstruction occurred, even in recanalized capillaries (*Figure 5A*). Consistent with the hypothesis that VEGF-R2 signaling is sensitive to changes in blood flow (*27*), obstructed capillaries (30 min after microsphere injection) had significantly lower colocalization with pVEGF-R2-labeled capillaries [sham 13.41 ± 2.4% compared to obstructed 3.85 ± 1.3%, unpaired t-test $t_{(30)}$ = 3.86, p=0.0006, $n_{(sham)}$=3 mice, 16 vessels, $n_{(obstructed)}$=4 mice, 16 vessels]. Three hours after microsphere injection when capillaries have begun to recanalize, we identified two distinct populations of recanalized capillaries based on pVEGF-R2 expression co-localization with DiI-labeled endothelium (*Figure 5B*, insert). Recanalized capillaries exhibited either significantly lower or higher pVEGF-R2 expression, than capillaries that were still obstructed by microspheres or unobstructed control capillaries (*Figure 5B*).

Although immunohistochemical analysis clearly indicated that VEGF-R2 signaling was altered in recanalized capillaries, its precise role remained ambiguous. Therefore, we utilized gain of function and knockdown experiments to better define the role of VEGF-R2 signaling in capillary recanalization. First, we stimulated VEGF-R2 signaling using our previously validated protocol (*Taylor et al., 2015*) of injecting 25 ng VEGF-A (i.c.v.) just prior to injection of microspheres. VEGF treatment significantly increased obstruction density 24 hr after microsphere injection compared to vehicle-injected mice (*Figure 5C*, unpaired t-test, $t_{(8)}$=3.282, p=0.01, only cortex contralateral to injection site was analyzed). Based on the fact that reduced levels of vascular pVEGF-R2 correlated with

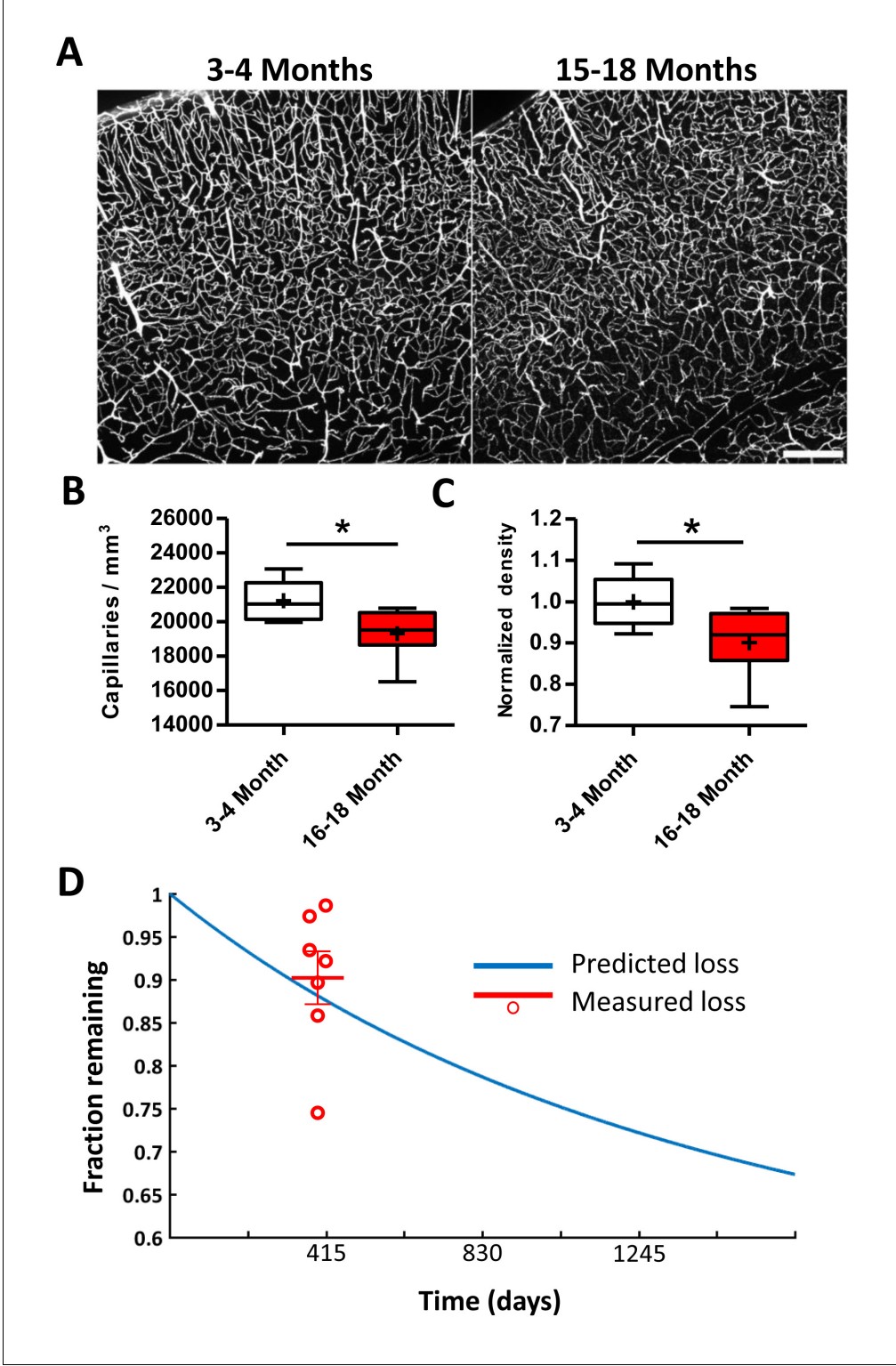

**Figure 4.** Lower capillary density in aged mice is predicted by obstruction and pruning rates. (**A**) Representative confocal images of Evans blue filled blood vessels in somatosensory cortex of young (3–4 month) and aged (15–18 month) mice. Scale bar 200 µm. (**B**) Box and whisker plot of capillary density (+denotes mean) across all cortical layers for 3–4 and 15–18 month old *Tek*-GFP mice (n = 6–7 mice; unpaired $t_{(11)}$=2.47, p=0.03). Error bars are S.E.M. (**C**) Box and whisker plot of normalized cortical capillary density for 3–4 or 15–18 month old *Tek*-GFP mice ($n_{(3-4month)}$=6, $n_{(16-18month)}$=7 mice, unpaired $t_{(11)}$=2.47, p=0.03). Error bars are S.E.M. (**D**) Predicted capillary loss over

*Figure 4 continued on next page*

*Figure 4 continued*

time (blue line) based on measured rates of spontaneous obstructions and pruning (see Materials and methods, *Figure 4—figure supplement 1*). Red data points and mean (±SEM) represent measured capillary loss in 15–18 month old aged mice normalized to 3–4 month old mice. Predicted capillary loss closely matched experimentally measured loss in aged mice (unpaired t-test, p=0.52).

DOI: https://doi.org/10.7554/eLife.33670.016

The following figure supplement is available for figure 4:

**Figure supplement 1.** Modeling capillary loss over time.

DOI: https://doi.org/10.7554/eLife.33670.017

recanalization in some but not all capillaries, and that increasing VEGF-R2 signaling lowered recanalization rates, we then asked whether reducing VEGF-R2 signaling could improve recanalization rates. For this we employed two approaches (*Figure 5D*), an endothelial-specific inducible knockdown of VEGF-R2 signalling (*Tek* Cre-ER$^{T2}$ crossed with floxed Kdr (VEGF-R2) line) or pharmacological inhibition of VEGF-R2 with the blood brain barrier permeable, small molecule inhibitor SU5416 (*Fong et al., 1999*; *Mendel et al., 2000*; ). Both of these approaches significantly reduced VEGF-R2 signaling as indicated by lower immunohistochemical staining for pVEGF-R2 [*Figure 5E*; tamoxifen vs vehicle: unpaired t-test, $t_{(6)}$=3.32, p=0.01; SU5416 vs vehicle: unpaired t-test $t_{(6)}$=3.17, p=0.02] or western blot detection of VEGF-R2 protein levels in the cortex (*Figure 5—figure supplement 1A*).

The impact of reduced VEGF-R2 signaling on capillary recanalization was assessed with in vivo time lapse imaging (*Figure 5F,H*) or post-mortem estimates of microsphere density at 4 and 21 days after microsphere injection (*Figure 5G,I*). For both types of analysis, rates of capillary recanalization were significantly enhanced when VEGF-R2 signaling was reduced with VEGF-R2 knockdown (*Figure 5F,G*) or pharmacological inhibition (*Figure 5H, I*). Not surprisingly, improved recanalization was associated with a significant reduction in the number of pruned capillaries at 21 days (*Figure 5F,H*). The effects of VEGF-R2 knockdown or inhibition on recanalization rates were not explained by any global cardiovascular changes, as we found no differences in heart rate, breath rate, blood pressure or tissue oxygenation between treated and control groups (*Figure 5—figure supplements 1–2*). Further, these effects could not be explained by significant differences in the initial density of microspheres injected [microsphere density 30-min post-injection: vehicle vs. SU5416 = 652 ± 224 versus 562 ± 200 microspheres/mm$^3$, $t_{(6)}$ =0.25, p=0.8; vehicle vs Tamoxifen: 509 ± 234 versus 328 ± 70 microspheres/mm$^3$, $t_{(10)}$= 2.0, p=0.07]. Lastly, VEGF-R2 knockdown or inhibition did not significantly alter RBC velocity in either control, recanalized or adjacent vessels (*Figure 5—figure supplements 1–2*). In conclusion, these experiments indicate that VEGF-R2 signaling plays a critical role in dictating capillary recanalization.

## Discussion

Here, we used in vivo two-photon imaging to characterize the inherent risk of cortical capillary obstruction in adult mice. We found that on average ~1 in 10,000 capillaries will become obstructed for longer than 20 min within a 2 hr window. However, the risk of obstruction was not equally distributed as superficial capillaries of lower arteriole branch orders (distribution peaks at the 3rd arteriole branch order) were at highest risk (*Figure 1C*). While this distribution of obstructed capillaries (as a function of arteriole branch order) bears similarity to a recent study (*Erdener et al., 2017*), we did not quantify risk relative to venous branch order which may have a higher risk for obstruction (*Erdener et al., 2017*). The absolute number of obstructed cortical capillaries might seem quite small; however, it is important to keep in mind that the mouse cortex occupies ~180 mm$^3$ (*Badea et al., 2007*) with an estimated 3.6 million capillaries (*Figure 4*) (; *Schmid et al., 2017*19, 33). Therefore, at any given time, one could expect a few hundred obstructed capillaries in the mouse cortex. We would argue that over time, this small fraction of obstructed capillaries could significantly impact cerebrovascular density and blood flow. For example, our data show that ~30% of obstructed capillaries were eventually pruned with no compensatory sprouting. Over a year, we would predict a 12% reduction in capillary density which closely approximates our experimentally determined estimate of vessel loss, as well as others (; *Hinds and McNelly, 1982*; ; *Buchweitz-Milton and Weiss, 1987*; 11-14, 34). In addition to these anatomical changes, we also show that capillary pruning locally

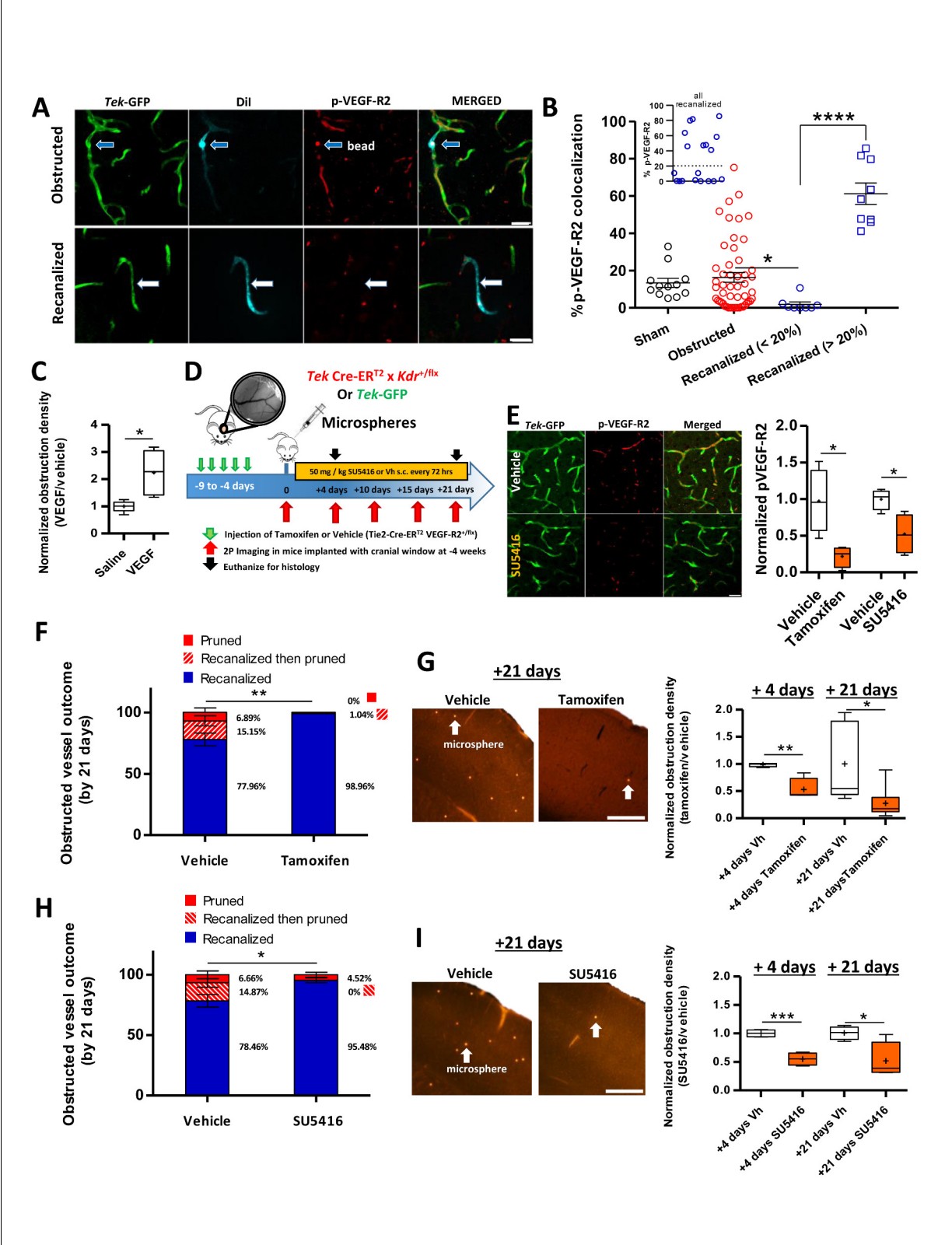

**Figure 5.** VEGF-R2 signaling dictates capillary recanalization. (**A**) Confocal images show phosphorylated VEGF-R2 (Y1175) immunolabeling in an obstructed capillary (top row, see DiI-coated microsphere) and one that recanalized (bottom row, note DiI labeling of GFP-labeled endothelium without presence of microsphere). (**B**) Histogram shows % pVEGF-R2 colocalization with endothelium at the site of an obstructed or recanalized capillary 3 hr after microsphere injection. Sham capillaries were measured from mice injected with vehicle solution but without microspheres. Inset shows the

*Figure 5 continued on next page*

*Figure 5 continued*

distribution of pVEGF-R2 colocalization values for all recanalized capillaries. Based on this distribution, we used a 20% cut off (dotted line) to separate the two distinct populations. Average eight capillaries per mouse, range 5–11. Note that recanalized capillaries exhibit significantly higher or lower pVEGF-R2 colocalization. n = 83 vessels total with 3 sham and seven injected mice, one way ANOVA $F_{(3,77)}$=22.93, p<0.0001, unpaired t-test to compare groups. *p<0.05, ****p<0.0001. (C) Box and whisker plots (+denotes mean) show normalized density of microspheres in the cortex of saline and VEGF injected (i.c.v.) mice 24 hr after microsphere injection. n = 5 mice per group, unpaired t-test. *p<0.05. Microsphere density: Vehicle 7.97 ± 1.3 per $mm^3$ versus VEGF injected 22.27 ± 2.1 per $mm^3$. (D) Summary and timeline of VEGF-R2 knockdown or inhibition experiments. E) Left: Immunolabeling for pVEGF-R2 in *Tek*-GFP mice shows reduced vascular expression 3 hr after injection of 50 mg/kg SU5416. Right: Quantification of vascular pVEGF-R2 in Tamoxifen-treated *Tek*CreER$^{T2}$ X $Kdr^{+/fl}$ mice [four mice per group; unpaired t-test, $t_{(6)}$=3.32, p=0.01] or *Tek*-GFP mice injected with SU5416 [four mice per group; unpaired t-test $t_{(6)}$=3.17, p=0.02]. (F) Capillary fates 21 days after obstruction in vehicle or tamoxifen treated *Tek* CreER$^{T2}$ X $Kdr^{+/fl}$ mice based on in vivo time lapse imaging (n = 6 mice per group, vehicle = 75 capillaries, Tamoxifen = 81 capillaries, Average 10 capillaries per mouse, range 5–16). Unpaired t-test to compare % recanalized, $t_{(10)}$=3.88, p=0.003.**p<0.01. G) Left: Representative images of cortical microspheres in coronal brain sections from vehicle or Tamoxifen-injected *Tek* CreER$^{T2}$ X $Kdr^{+/fl}$ mice 21 days after microsphere injection. Scale bar 500 μm. Right: Normalized density of microspheres in the cortex of vehicle and Tamoxifen-injected mice at 4 and 21 days after microsphere injection (n = 4–9 mice; unpaired t-test at 4 days $t_{(6)}$=4.43, p=0.004; and 21 days $t_{(12)}$=2.74, p=0.017.+is mean. *p<0.05, **p<0.01). Microsphere densities (/$mm^3$) at +4 days: Vehicle 16.2 ± 1.5, Tamoxifen injected 10.3 ± 5, +21 days: Vehicle 6.38 ± 3, Tamoxifen injected 1.5 ± 0.4. (H) In vivo determination of capillary fates after 21 days in vehicle or SU5416 injected *Tek*-GFP mice [n = 4 mice per group at 4 days, n = 5 for vehicle and n = 9 for tamoxifen injected at 21 days, vehicle = 84 capillaries; SU5416 = 90 capillaries; Average 20 capillaries per mouse, range 16–30, unpaired $t_{(7)}$=2.73, p=0.03]. I) Left: Images show cortical microspheres in brain sections from vehicle and SU5416 treated *Tek*-GFP mice 21 days after microsphere injection. Scale bar 500 μm. Right: Normalized density of microspheres in the cortex of vehicle or SU5416 injected mice at 4 and 21 days [n = 4 mice per group; unpaired t-test at 4 days $t_{(6)}$=6.97, p=0.004, and 21 days $t_{(6)}$=2.92, p=0.02]. Microsphere densities (/$mm^3$) at +4 days: Vehicle 69.2 ± 1.4, SU5416 injected 31.94 ± 4.2, +21 days: Vehicle 21.59 ± 5.3, SU5416 injected 12.21 ± 1.33. *p<0.05, **p<0.01, and ***p<0.001 compared to vehicle injected. Error bars are S.E.M.

DOI: https://doi.org/10.7554/eLife.33670.018

The following figure supplements are available for figure 5:

**Figure supplement 1.** Vascular-specific knockdown of VEGF-R2 does not affect cardiovascular health or blood flow.
DOI: https://doi.org/10.7554/eLife.33670.019

**Figure supplement 2.** Inhibition of VEGF-R2 signaling with SU5416 did not affect cardiovascular health or blood flow.
DOI: https://doi.org/10.7554/eLife.33670.020

perturbs blood flow and flux in adjacent capillary segments for several days. This supports previous experimental work (*Schaffer et al., 2006*) showing that capillary occlusion alters blood flow in branches up and downstream of the occlusion. Other work has shown through modelling that micro-vessel loss can lead to abnormalities in local perfusion (). Thus, despite the highly interconnected nature of the capillary bed, even a small number of pruning events can significantly alter local blood flow and capillary density.

## Microsphere model of obstruction

Previous studies have injected relatively large (10–100 μm) microspheres in adult rodents to model multiple small ischemic strokes (; ; *Rapp et al., 2008*; *Roos and Ericsson, 1999*; ) or to identify novel mechanism of vessel recanalization (*Lam et al., 2010*; *Grutzendler et al., 2014*). Here we used small 4 μm diameter microspheres to selectively obstruct capillaries in an attempt to model spontaneous, naturally occurring capillary obstructions in the brain. Taking this approach provided several experimental advantages. Due to their small size, microspheres can be injected through the tail vein instead of through more invasive routes, such as the internal carotid artery which is believed to be required for emboli larger than 10 μm to reach the brain. The non-invasive nature of these injections facilitates repeated injections and minimizes exposure to anesthetics needed for injections into the carotid. The small size of the microspheres allows them to be dispersed throughout the brain. While we did not find any differences in the anterior to posterior distribution of microspheres in dorsal cortex (from −2 mm to +1.5 mm relative to bregma), this does not take into account regional differences that may exist in more anterior, posterior and lateral cortical regions. Since regional vulnerability to capillary obstruction has relevance to pathophysiology, future work will be needed to characterize the risk of obstruction and pruning across the cortical mantle or subcortical regions. We have also exhaustively validated this model of obstruction. First, we have established that our dosage of microsphere injection did not cause detectable changes in global cardiovascular parameters and blood chemistry (*Figure 1—figure supplement 2*). Thus, while microspheres do obstruct vessels in other organs, they do not induce cell death, or organ dysfunction. A future question will be to study

capillary obstructions in other organ systems to see how the risk of obstructions, and ultimate fates vary in different systems. Angiophagy for example was first described in brain vasculature but turned out to be a universal property of most vessels (*Grutzendler et al., 2014*). Second, how the brain deals with microsphere obstructions yielded similarities with those that occur spontaneously. For example, rates of capillary recanalization and eventual pruning of obstructed capillaries were similar between the two types of obstructions (microsphere vs natural emboli). A third advantage is that microspheres can be coated with lipophilic dye DiI, thereby allowing us to tag obstructed vessels, even after they had recanalized. This method will be useful for furthering our understanding of the signaling events that occur and differentiate recanalized capillaries versus those that remain obstructed.

However, there are important caveats that should be noted with the microsphere model. Foremost is the fact that the composition (polystyrene microsphere vs. fibrin, blood cells) and size of the obstruction likely affects the route of recanalization. While we found low rates of angiophagy with microsphere obstructions, this likely would not be the case with larger and more natural emboli (such as cholesterol or fibrin-based clots). For example, Lam et al. showed different rates of washout versus angiophagy between natural emboli compared to microspheres (*Lam et al., 2010*). A second caveat that could not be avoided was that we were unable to inject microspheres without the use of anesthesia. Anesthetics such as isoflurane have pleiotropic effects that influence synaptic transmission, blood flow and ischemic cell death (*Santisakultarm et al., 2016*; *Yamakura and Harris, 2000*; *Seto et al., 2014*). Therefore, we cannot completely rule out a possible effect of even brief isoflurane exposure (in awake imaging or histology experiments) on microsphere obstruction density, as others have reported anesthetic effects on capillary stalling (*Erdener et al., 2017*; *Santisakultarm et al., 2016*).

## Vascular remodeling in the mature brain

Our finding that a failure to remove obstructions inevitably leads to capillary pruning suggests that the accumulation of obstructions over time is at least partially responsible for the loss of microvascular density in aging reported by us and others (*Figure 4*) (*Klein and Michel, 1977*; *Hinds and McNelly, 1982*; ; *Buchweitz-Milton and Weiss, 1987*; ; *Amenta et al., 1995a*; *Amenta et al., 1995b*; *Harb et al., 2013*) [see (*4Riddle et al., 2003*) for review, but also (; *Hughes and Lantos, 1987*) for studies failing to find age related losses in capillaries)]. The fact that vessel pruning was not compensated for with sprouting (angiogenesis) is significant, but should not be viewed as completely surprising since other in vivo longitudinal imaging studies of cortical microcirculation have found scant evidence for angiogenesis (at least below the cortical surface) in the mature brain (*Reeson et al., 2015*; ; *Tennant and Brown, 2013*; 31, 34, 49, 50). Furthermore, since our obstructions targeted a small fraction of capillaries and therefore did not lead to hypoxia or cell death, the hypoxia-related signals needed to trigger angiogenesis would not be present.

Our estimate of capillary loss in the aged mouse cortex, while closely matching our predicted loss and many other studies, would seem conservative to some prior reports (*Figure 4—figure supplement 1*). Discrepancies in empirical estimates of capillary loss with age may reflect different methodologies for visualizing capillaries. Our measurements were based on intravenous injection of fluorescent dyes therefore labeling flowing capillaries in the brain with very high signal to noise. However, this could slightly overestimate the density of patent capillaries since injected plasma dyes also filled stalled capillaries given that we never found a *Tek*-GFP-labeled capillary without fluorescent dye present in it. Importantly, our Evans-blue-based estimates of vessel density in young adult mice closely matches other published studies (; *Schmid et al., 2017*20, 33), particularly those by the Kleinfeld lab (; *Tsai et al., 2009*). In contrast, many previous studies in rats have used histochemical staining for alkaline phosphatase (*Jucker et al., 1990*; *Amenta et al., 1995a*; ; *Amenta et al., 1995b*15, 16, 23, 24) or electron microscopy (*Casey and Feldman, 1985*) to estimate vessel density. The caveat of using alkaline phosphatase staining is that it assumes ubiquitous vascular expression that does not vary with age (independent of vessel loss), while electron microscopy studies are constrained by very low sample sizes. Another obvious factor underlying discrepancies in capillary loss could be the experimental animal studied since most previous studies used rats. While we are confident that our predictions are well suited for mice, it would be interesting to know if it could be applied to other larger and longer-lived species. Future studies, perhaps replicating our approaches

for estimating capillary obstruction and clearance rates, as well as capillary density in aging animals would be necessary to extend our predictions.

## Mechanisms of recanalization

Historically, the main mechanism of recanalizing a cerebral microvessel was believed to be through Tissue Plasminogen Activator (TPA), which was provided by microvascular endothelial cells (*del Zoppo and Mabuchi, 2003*). However, previous work has shown in larger vessels the window for TPA-mediated recanalization is short (; *Grutzendler et al., 2014*42, 43), and many types of obstructions are not susceptible to thrombolytics. The recent discovery of angiophagy (; *Grutzendler et al., 2014*42, 43) in penetrating arterioles and capillaries suggested different strategies exist for clearing obstructions. Our finding of an angiophagy-like process where endothelial cells engulf and extrude emboli through the vessel wall replicates previous findings of the Grutzendler lab (; *Grutzendler et al., 2014*42, 43). However, we cannot rule out the possibility that leukocytes (which were not labelled in our study) could have engulfed microspheres and transported them across the capillary wall. Due to the low incidence (2%) of angiophagy-like events in our microsphere model, we could not extensively characterize this process. Further, the manner and frequency of angiophagy-like events is likely influenced by the specific size of emboli and its composition. For example, larger occlusions (>4 μm) would have more severe consequences such as hypoxia and cell death (*Hughes and Lantos, 1987*) which can trigger major structural changes in the vascular endothelium through hypoxia-inducible factors. An intriguing question that has not been resolved is what mechanisms mediate the dislodging and extrusion of obstructions back into the circulation. Although we know that VEGF-R2 signalling plays a critical role in recanalization, we do not know the precise mechanisms through which it acts. For example, VEGF-R2 is coupled to nitric oxide and other signaling pathways (*27*) that could mediate a change in vessel tone or diameter in the obstructed capillary. However, a recent study has questioned the role of NO as a major capillary dilator (*52*), thus other pathways may be involved. We should also note that heamodynamic forces from up or downstream capillaries are not likely a critical factor since measured levels of blood flow or diameter in these capillaries did not precede or predict capillary recanalization (*Figure 2—figure supplement 3*). Another possibility that will require further study is whether pericytes facilitate recanalization or pruning. In particular, smooth muscle actin-positive pericytes, which are found at precapillary arterioles and lower branch order capillaries, have been shown to regulate capillary flow (*Hill et al., 2015*). Furthermore, ischemic stroke leads to prolonged pericyte constriction which can limit blood flow in capillaries (*Hill et al., 2015*; *Hall et al., 2014*53, 54). Therefore, it is possible that pericytes may contribute to capillary recanalization by regulating microvessel tone or vasomotion. Pericytes could also play a significant role in vessel pruning since they are hotspots of matrix metalloprotease activity associated with BBB disruption, (*Underly et al., 2017*) which is needed for remodeling of the extracellular matrix and basement membrane around capillaries. The only variable that differentiated recanalization successes versus failure was distance to the nearest flowing vessel, which suggest at least a more complex interplay between vessel tone and heamodynamic forces. The precise mechanisms by which vessel tone and blood flow intersect to dislodge and extrude persistent emboli will require further study.

## Microvascular obstructions and cognitive impairment

Although not the focus of the present study, it is very likely that the incidence of capillary obstructions and the rate of recanalization and pruning could be strongly influenced by disease states. Indeed, a recent study (*Santisakultarm et al., 2014*) showed that rates of capillary obstruction were significantly elevated in blood disorders with excessive production of platelets and RBCs. It is now accepted that there is a strong vascular component to many types of dementia (; *Tong and Hamel, 2015*3, 17). While the umbrella of Vascular Cognitive Impairment (VCI) encompasses many different vascular pathologies, microvessel loss is an important one (; *Riddle et al., 2003*; *Tong and Hamel, 2015*; *Langdon et al., 2017*). Our study offers at least one mechanism for the loss of microvessels in the healthy aging brain, therefore it is tempting to speculate that similar mechanisms could be at work for known risk factors of dementia such as diabetes or hyperlipidemia. To our knowledge, no study has tracked capillary obstructions and their long-term fates in animal models of VCI. In this sense, we anticipate our study will provide a useful framework for understanding microcirculatory

changes that accompany, and possibly underlie cognitive impairment. Furthermore, since we have shown that VEGF-R2 plays a critical role in dictating capillary recanalization, future studies could apply this knowledge to ameliorate disturbances in brain circulation and function/cognition in neurological conditions with a known vascular connection.

# Materials and methods

## Key resources table

| Reagent type (species) or resource | Designation | Source or reference | Identifiers | Additional information |
|---|---|---|---|---|
| Gene | | NA | | |
| Strain, strain background (*Mus musculus*) | Tek-GFP | The Jackson Laboratory | Tie2-GFP_JacksonLab:003658 (IMSR Cat# JAX:003658, RRID:IMSR_JAX:003658) | 2–4 month-old males, and 16–18 month old males and females used |
| Strain, strain background (*M. musculus*) | Kdr$^{fl/fl}$ | Drs. Jane Rossant and Andras Nagy (*Forde et al., 2002*) | | 2–4 month-old males |
| Strain, strain background (*M. musculus*) | Tek-CreER$^{T2}$ | EMMA | Tie2-CreER$^{T2}$_EMMA:00715 | 2–4 month-old males |
| Genetic reagent (*M. musculus*) | Tek-CreER$^{T2}$ X Kdr$^{+/fl}$ | in house breeding (*Mendel et al., 2000*) | | Tek-CreER$^{T2}$ mice (*Langdon et al., 2017*) (EMMA 00715) bred with Kdr$^{fl/fl}$ line (*Forde et al., 2002*) |
| Cell line | | | | |
| Transfected construct | | | | |
| Biological sample | | | | |
| Antibody | pVEGF-R2 | Cell Signaling | 19A10 Rabbit mAb #2478 | (1:300) For IHC |
| Antibody | Cy5 conjugated secondary | Invitrogen | A10523 | (1:400) For IHC |
| Antibody | VEGF-R2 | Cell Signaling | CS2479s | (1:1000) For WB |
| Antibody | β-actin | Sigma | A-5441 | (1:2000) For WB |
| Antibody | Goat anti-rabbit IgG-HRP | Cell Signaling | 7074S | (1:1000) For WB |
| Antibody | goat anti-mouse IgG-HRP | KPL | (*Riddle et al., 2003*; *Kleinfeld et al., 1998*; *Santisakultarm et al., 2014*; *Villringer et al., 1994*; *Erdener et al., 2017*; *Gould et al., 2017*; *Mann et al., 1986*; *Klein and Michel, 1977*; *Hinds and McNelly, 1982*; *Casey and Feldman, 1985*; *Buchweitz-Milton and Weiss, 1987*; *Jucker et al., 1990*; *Amenta et al., 1995a*; *Tong and Hamel, 2015*; *Reeson, 2018*) | (1:1000) For WB |
| Recombinant DNA reagent | | | | |
| Sequence-based reagent | floxed *Kdr* Wildtype | Integrated DNA Technologies (IDT) | 5'-TGG-AGA-GCA-AGG-CGC-TGC-TAG-C-3' | 30 µM |
| Sequence-based reagent | floxed *Kdr* Flx | Integrated DNA Technologies (IDT) | 5'-CCC-CCT-GAA-CCT-GAA-ACA-TA-3' | 30 µM |

*Continued on next page*

*Continued*

| Reagent type (species) or resource | Designation | Source or reference | Identifiers | Additional information |
|---|---|---|---|---|
| Sequence-based reagent | floxed *Kdr* Common | Integrated DNA Technologies (IDT) | 5'-CTT-TCC-ACT-CCT-GCC-TAC-CTA-G-3' | 30 µM |
| Sequence-based reagent | Cre Fwd | Integrated DNA Technologies (IDT) | 5'-CGA-GTG-ATG-AGG-TTC-GCA-AG-3' | 30 µM |
| Sequence-based reagent | Cre Rev | Integrated DNA Technologies (IDT) | 5'-TGA-GTG-AAC-GAA-CCT-GGT-CG-3' | 30 µM |
| Peptide, recombinant protein | VEGF165 protein | Sigma | V4512 | 1 µL of 25 ng in artificial cerebrospinal fluid |
| Commercial assay or kit | BCA protein assay | Pierce | #23225 | 562 nm absorbance |
| Chemical compound, drug | BioRad Clarity Western ECL Substrate | BioRad | Cat# 107–5061 | |
| Chemical compound, drug | microspheres (4 µm diameter) | Life Technologies | F8858 | 2% solids; peak emission 605 nm |
| Chemical compound, drug | Rhodamine B dextran | Sigma | R9379 | 4%, average molecular weight 70,000, in 0.9% saline |
| Chemical compound, drug | FITC dextran | Sigma | 46945 | 4%, average molecular weight 70,000, in 0.9% saline |
| Chemical compound, drug | Evans blue | Sigma | E2129 | 2% in 0.9% saline |
| Chemical compound, drug | Tamoxifen | Sigma | T5648 | five daily intraperitoneal injections of 125 mg/kg dissolved in corn oil, Sigma, C8267 |
| Chemical compound, drug | SU5416 | Tocris | Cat# 3037 | 50 mg/kg, 0.5% w/v carboxy methyl cellulose, 0.9% sodium chloride, 0.4% polysorbate, 0.9% benzyl alcohol in dH2O |
| Chemical compound, drug | Hoechst 33342 | Thermoscientific | CA#62249 | (1/10,000) |
| Chemical compound, drug | DiI | Thermoscientific | CA#D282 | 30 mg DiI dissolved in 5 mL 100% EtOH |
| Software, algorithm | Fluoview | Olympus | FV10-ASW 4.2 | |
| Software, algorithm | FIJI | Max Planck Institute of Molecular Cell Biology and Genetics 58) | (Fiji, RRID:SCR_002285) | |
| Software, algorithm | ImageJ | Wayne Rasband, NIH USA | ImageJ 1.47 v (National Center for Microscopy and Imaging Research: ImageJ Mosaic Plug-ins, RRID:SCR_001935) | |
| Software, algorithm | MATLAB | Mathworks | R2017a (MATLAB, RRID:SCR_001622) | Institutional Licence |
| Software, algorithm | VesselNumEst | This paper | Source code 1 | Runs as macro in ImageJ, available at https://github.com/elifesciences-publications/eLIFE- |
| Software, algorithm | Capillary Loss Modeling | This paper | Source Code 2 | Runs in MATLAB with Database S1 as input, archived at https://github.com/elifesciences- |
| Other | | | | |

## Animals

We used 2–4 month, and 16–18 month old *Tek*-GFP mice (The Jackson Laboratory, 003658), or *Tek*-CreER$^{T2}$ mice (*Forde et al., 2002*) (EMMA 00715) bred with *Kdr*$^{fl/fl}$ line (*Hooper et al., 2009*) to generate *Tek*-CreER$^{T2}$ X *Kdr*$^{+/fl}$ mice. Male mice were used in all experiments except for determining capillary density in 3–4 versus 16–18 month mice (*Figure 4*), in which retired breeders (male and female) were compared to gender matched 3–4 month-old *Tek*-GFP mice. No differences in capillary density were found by sex, so data were pooled (3–4 month male versus female Unpaired $t_{(4)}$=0.69, p=0.52; 16–18 month males versus females Unpaired $t_{(5)}$=1.36, p=0.23). Offspring were genotyped using the following primers for floxed *Kdr* (WT: 5'-TGG-AGA-GCA-AGG-CGC-TGC-TAG-C-3' and Flx: 5'-CCC-CCT-GAA-CCT-GAA-ACA-TA-3', and common: 5'-CTT-TCC-ACT-CCT-GCC-TAC-CTA-G-3') and Cre (5'-CGA-GTG-ATG-AGG-TTC-GCA-AG-3' and 5'-TGA-GTG-AAC-GAA-CCT-GGT-CG-3'). Mice were housed under 12 hr light/dark cycle and given *ad libitum* access to water and laboratory diet. All experiments were conducted according to the guidelines set by the Canadian Council of Animal Care, ARRIVE, and approved by the University of Victoria Animal Care Committee, protocol 2016-016.

## Cardiovascular measurements

Blood pressure was measured using a commercially available system (Kent Scientific Mouse CODA). Mice were lightly anesthetized (1% isoflurane) and placed on a heating pad with body temperature maintained at 37°C. Tissue oxygenation (% $O_2$ saturation), heart rate (beats/min), and breathing rate (breaths/min) were measured with the commercially available Starr Mouse Ox system (Starr Life Sciences Corp.). Values for each time point were taken as the average of at least 5 min of continuous recording. Blood chemistry was measured using Abaxis VetScan i-STAT system (with CG8 +cartridges). To obtain blood (~100 µL) for analysis, mice were anesthetised with isoflurane and rapidly decapitated.

## Cranial window surgeries

Mice were anesthetized with isoflurane (2% induction, 1.3% maintenance) in medical grade air and fitted into a custom-made surgical stage with body temperature maintained at 37°C. A 0.03 mL bolus of 2% dexamethasone was given intramuscularly to reduce inflammation associated with the procedure. Following a midline incision and retraction of the scalp, a custom metal ring (outer diameter 11.3 mm, inner 7.0 mm, height 1.5 mm) was affixed to the skull with cyanoacrylate glue. Using a high-speed dental drill, a 4 mm diameter craniotomy was made overlying the somatosensory cortex. Cold HEPES-buffered artificial cerebrospinal fluid (ACSF) was applied to the skull intermittently during the drilling procedure to keep the brain cool and reduce inflammation. The dura was left intact and a 5 mm coverslip (no. one thickness) was placed over the brain and secured to the surrounding skull with cyanoacrylate glue and dental cement. The surrounding skin was then secured to the edges of the metal ring with cyanoacrylate glue. Mice were allowed to recover under a heat lamp before being returned to their home cage. After 4 weeks of recovery, mice that showed significant loss of clarity to the imaging window were excluded from the study.

## Microsphere model of capillary obstruction

For inducing capillary obstructions, 20 µL of microspheres (4 µm diameter; 2% solids; peak emission 605 nm; Life Technologies FluoSpheres sulfate, F8858) were mixed with 100 µL of fluorescent dye or saline and injected in the tail vein. For comparing obstruction clearance rates between experimental groups (e.g. mice with treated VEGF-R2 inhibitor or vehicle), a master solution of microspheres was first made, sonicated (3 min) and then aliquoted into separate injection doses which were then assigned to mice randomly. Groups were always run in parallel and balanced between controls and experimental conditions.

## In vivo imaging

Mice were lightly anesthetized with 1% isoflurane with body temperature maintained at 37°C. Mice were given an intravenous injection of 4% Rhodamine B dextran (Sigma, R9379, average molecular weight 70,000, in 0.9% saline) or FITC dextran (Sigma, 46945, average molecular weight 70,000, in 0.9% saline). The cerebral vasculature was imaged through the cranial window using an Olympus

FV1000MPE multiphoton laser scanning microscope equipped with a mode-locked Ti:Sapphire laser (Spectral Physics) tuned to either 850 or 910 nm (for Rhodamine B or FITC dextran, respectively). Laser power delivered by each wavelength ranged from 15 to 50 mW at the back aperture depending on imaging depth. Images were acquired in a stack (2 µm z-step) with either a 40x Olympus IR-LUMPlanFl (NA = 0.8, 0.20 µm/pixel) or 20x Olympus XLUPlanFl water-immersion objective (NA = 0.95, 0.62 µm/pixel), using Olympus Fluoview FV10-ASW software. Emitted light was separated by a dichroic filter (552 nm) and then directed through a bandpass filter (either 495–540 nm or 558–706 nm).

For awake imaging, mice were habituated to imaging in three sessions; one session per day for 3–5 days before the start of the experiment. For each session, mice were very briefly anesthetized, then swaddled in a cotton wrap and restrained in a custom plastic rodent restraint device, modified to allow the head to be fixed in a custom stage for imaging. Mice were allowed to wake up in the restraint device. The duration of restraint was gradually increased from 5 to 30 min over the three habituation sessions. Any mouse that showed abnormal or prolonged signs of distress during habituation or imaging sessions were removed from the study (~25% of mice). In general, awake-imaging sessions lasted 20–60 min.

Imaging areas were chosen based on the clarity of the window and were located near the center of the cranial window, with penetrating arterioles (identified from surface pial vasculature) in the field of view. For determining the rate of spontaneous capillary obstructions that occur in a 2 hr period, multiple areas were randomly selected and imaged every 10–20 min (~50–350 µm below pial surface). For longitudinal imaging of spontaneous or microsphere-induced capillary obstructions, 2–4 regions were imaged based on suitable density of occlusions and imaging clarity. Any obstructed capillary with more than one microsphere lodged, or less than two branching points from another obstruction were excluded. In no cases did we observe larger vessels (>~4 µm) obstructed by aggregates of multiple microspheres.

## Analysis of vascular structure and flow

Blood flow velocity was estimated from a series of 3 line scans conducted on capillaries ($\leq$8 µm in diameter) with a 30 s waiting period between each scan. In each brain region, 2–3 microsphere-obstructed capillaries were selected for line scans (to confirm absence of blood flow) and 1–2 adjacent flowing segments. In all cases, 1–2 capillaries in each region that were both flowing and at least two branching points away from any obstruction were used as within animal control vessels. All analysis for blood flow velocity, vessel lumen diameter, and RBC flux was conducted by two separate independent researchers blinded to experimental condition. To measure RBC velocity, two blinded researchers chose three equally spaced apart RBC streaks from each line scan (each linescan was on average 30.83 ± 1 ms per scan, 2 µm/pixel) and using built in functions of Olympus Fluoview software measured the inverse slope ($\Delta$ time/ $\Delta$ distance). This was repeated for all three linescans done for a single vessel at each timepoint (30 s apart). Therefore, each single vessel RBC velocity at each timepoint was the average of 18 independent blinded measurements (two experimenters' x 3 linescans x 3 RBC slopes). Any curvilinear slopes were excluded. Capillary diameter was measured at Full Width at Half Maximum (FWHM) from a Gaussian fitted intensity profile (John Lim, IMB. March 2011) drawn perpendicular to the capillary. RBC Flux was calculated assuming laminar flow, as $F = \pi/8 \cdot v \cdot d^2$ (*Schaffer et al., 2006*). Direction of flow was determined from the direction of the slope of RBCs in line scans (upwards or downwards) at all time points. Branching orders were determined for capillaries branching off single penetrating arterioles in each region, and each capillary that could unambiguously be assigned a branch order was included. Obstructed branch orders were determined working backwards from the obstructed capillary to a penetrating arteriole that could be followed to the pial surface and identified as an arteriole.

## Estimation of capillary numbers

Capillary numbers were estimated using a custom written macro in either ImageJ or FIJI (*Schindelin et al., 2012*) (see *Figure 1—figure supplement 1*, Source code 1)(*Reeson, 2018*). Imaging stacks of fluorescently labeled vasculature (rhodamine or FITC dextran) were split into sub-stacks of 10 images (2 µm z step). Sub-stacks were each maximally projected in the z plane, and automatically thresholded using ImageJ function Triangle (*Zack et al., 1977*), which best identified vascular

signal. The total area (of each image) and percent vascular signal (number of pixels after thresholding) was measured after applying a median filter (radius one pixels) to eliminate speckling. The area and vascular signal were then converted from pixels to μm and from 2D to 3D manually. The fractional vascular volume (v/v) and total vascular volume (μm³) was then calculated. Thresholded substacks were skeletonized (*Arganda-Carreras et al., 2010*) to create single pixel linear segments and total vascular length was taken as the total number of skeletonized pixels. From vascular length and area measurements, average vessel width was calculated. Since capillaries constitute the majority of the cortical vasculature (*Blinder et al., 2013*), we estimated the total number of capillaries by dividing the total vascular volume (μm³) with the average volume of a single capillary, assuming an average radius of 2 μm and length of 75 μm (*Blinder et al., 2013*) (average capillary volume: 942.48 μm³). We should note that flattening sub stacks and then re-projecting across the volume overestimates vascular volume (square vessels rather than cylinders) and underestimates vascular length (loss of length in z direction). Therefore, to validate our estimates we first created a sample set of imaging data (four mice, two imaging areas per animal) and created two independent and blinded manual measurements of vascular length (tracing vasculature by hand in each sub-stack) and capillary number (manual counts). Automated estimates of vascular volume were in agreement with and published data (*Tsai et al., 2009*). Likewise, the automated estimate of capillary numbers agreed with manual counts (paired t test, $t_{(3)}=0.33$, $p=0.76$).

## Recanalization rates and capillary fates

For estimating recanalization rates within the first 48 hr (*Figure 2D*), 3–4 regions per mouse were imaged from time 0 (~30 min after injection of microspheres) and then 6, 12, 24 and 48 hr later. Each imaging session was limited to 40 min total with mice regaining consciousness and returned to their home cage between each session. Recanalization rates were calculated as % of remaining obstructions at each time point for each animal. For long-term assessment of capillary fates, obstructed capillaries (microsphere induced or spontaneous; awake or anesthetized) were imaged at time 0 and then at 2, 4, 10, 15 and 21 days with a small subset of mice imaged up to 31 days. All apparent instances of recanalization were confirmed with vessel line scans. Vessel pruning was directly determined in *Tek*-GFP mice by noting the retraction of GFP-labeled endothelium or indirectly in *Tek* CRE ER$^{T2}$ *Kdr*$^{+/flx}$ mice by the complete the loss of labeled plasma (FITC dextran) in a fully pruned segment.

## Aged capillary density measurements

Male and female *Tek*-GFP retired breeders (16–18 months old mice) and gender matched 3–4 month-old *Tek*-GFP mice were anesthetized with isoflurane and intravenously administered 100 μL of 2% Evans blue in 0.9% saline (Sigma, E2129). Evans blue was allowed to circulate for 30 min and then mice were decapitated, and the brain fixed in 4% paraformaldehyde (PFA) in 0.1M phosphate buffered saline (PBS) overnight at 4°C. Brains were sectioned at 100 μm on a Leica vibratome (T1000) and every 3rd section was immediately mounted on charged slide and coverslipped with Fluoromount G (ThermoFisher, 00-4958-02). Aged and young mouse brains were processed and imaged in parallel. All sections were imaged immediately after cutting to minimize the possible leaching out of Evans blue dye that can occur over a 24 hr period. Evans blue was excited using a 635 nm laser and collected using an Olympus confocal microscope with a 10x objective (NA 0.40) and a Cy5 emitter filter (670–720 nm). Confocal image stacks were collected in 2 μm z-steps at a pixel resolution of 1.035 μm/pixel. All imaging parameters, including gain and offset, were kept consistent throughout. Three images of the somatosensory cortex were captured for each animal. For manual capillary counts, two independent blinded researchers z projected (maximal intensity) the middle 24 μm of each stack to minimize capillaries crossing over each other in the projection. Fluorescence levels between different animals were normalized by setting the scale of each image to 40% of the maximum pixel intensity of the brightest vessels. For a segment to be counted as a capillary, it must have been clearly visible and project at least 5 μm from another segment. Vessels with diameters greater than 8 μm were excluded.

## Modeling capillary loss over time

Capillary loss over time was modelled by custom written software in Matlab R2017a. Distribution of branch orders measured experimentally was applied to a theoretical set of 100,000 capillaries (without structural relationships between them). The risk of obstruction was (Source code 2)(*Reeson, 2018*), applied to each branch order based on the distribution of obstructed capillaries by branch orders (sum adding to observed rate). For each 2 hr cycle, the number of capillaries in each branch order was multiplied by its assigned risk to give the number of obstructed capillaries in each branch order. The number of obstructed capillaries was then multiplied by the overall risk of pruning (0.30) to calculate the number of pruned capillaries for each branch order, which was then subtracted from the number of vessels in each branch order category. Importantly, the risk of obstruction for each branch order was assumed constant (overall risk, therefore only changed as the distribution of branching orders varied). Individual capillary branch orders (initially assigned) were also fixed and not adjusted by any 'up stream' pruning events (capillaries were modeled without structural relationships between them).

## Stimulating or blocking VEGF signaling in vivo

Cage littermates were randomly selected to either control or VEGF-R2 knockdown/inhibition experimental groups. Cre recombinase activity was induced before microsphere or sham injection by five daily intraperitoneal injections of 125 mg/kg of tamoxifen (Sigma, T5648, dissolved in corn oil, Sigma, C8267), while vehicle injected mice received corn oil alone (*Reeson et al., 2015*; *Monvoisin et al., 2006*; *Sörensen et al., 2009*). In *Tek*-GFP mice, VEGF-R2 activity was inhibited by sub-cutaneous injections of SU5416 every 72 hr (*Fong et al., 1999*; *Mendel et al., 2000*) (Tocris, 50 mg/kg, 0.5% w/v carboxy methyl cellulose, 0.9% sodium chloride, 0.4% polysorbate, 0.9% benzyl alcohol in dH$_2$O) intiated 30 min after microsphere injection.

For stimulating VEGF signaling in the brain, mice were injected with 1 µL of 25 ng of recombinant mouse VEGF165 protein (Sigma V4512) in artificial cerebrospinal fluid (ACSF) or ACSF alone into the left lateral ventricle (2 mm lateral, 0.5 mm posterior of bregma). Twenty minutes after i.c.v injections, 20 µL of microspheres in 100 µL of 0.9% saline was injected into the tail vein. One day after microsphere injection, mice were killed, brains were extracted and fixed overnight in preparation for microsphere density analysis. Only the right hemisphere was used for analysis to avoid confounding effects of the injection site.

## Microsphere density analysis

Coronal brain sections (100 µm thick) from microsphere injected mice were imaged on an Olympus BX51 microscope with a 4x UPlanFLN objective (N.A. 0.13, 456.69 pixels/mm, 1.15 × 0.87 mm) using a Cy3 filter set on an Olympus DP73 digital camera using CellSens software. Images were taken of (every 3[rd] section) from the most anterior and medial sections of the cortex from +1.70 mm to −2.70 mm from bregma (*Franklin and Paxinos, 2008*). An experimenter blinded to condition counted the number of microspheres within cortical regions of interest to estimate microsphere density. To limit variability in total numbers of microspheres injected, mice were run in balanced groups of control and experimental conditions and received injections from the same diluted stock of microspheres. To compare between cohorts each experimental density of a cohort was normalized to the average density of that cohort's control animals. Normalized ratios were then averaged across cohorts (typically each experiment consisted of 2–4 cohorts).

## Analysis of endothelial cell regression

To determine endothelial cell (EC) density around pruned vessels, brains of *Tek*-GFP mice (21 days after microsphere or saline injection) were immersion fixed overnight in 4% PFA at 4°C. Brains were then sectioned at 100 µm on a vibratome. Every 3[rd] section was incubated in Hoechst 33342 (20mM stock, 1:10,000 dilution, Thermoscientific 62249) in 0.1M PBS for 20 min, washed and mounted on charged slides. Hoechst, GFP and fluorescent microspheres were sequentially excited using 405, 488 and 543 nm laser lines, respectively. Sections were imaged using an Olympus confocal microscope with a 20x objective (NA 0.75). Image stacks were collected in 2 µm z-steps at a pixel resolution of 0.31 µm/pixel. An EC was identified and included in the analysis if the Hoechst positive nucleus

showed complete colocalization with endothelial GFP signal. All EC within a 100 µm radius of the microsphere were counted.

## DiI coating of microspheres

To coat microspheres (1 or 4 µm diameter), an equal volume of microsphere stock solution was added to DiI solution (30 mg DiI dissolved in 5 mL 100% EtOH) and sonicated for ~1 hr and stirred overnight. The ethanol was then allowed to evaporate overnight at 37°C and the DiI-coated beads were reconstituted in 0.9% saline. Solution was further sonicated for ~1 hr to disperse microspheres. Sham injected controls had an equivalent solution of DiI (no microspheres) evaporated and reconstituted in saline.

## Phosphorylated VEGF-R2 immunohistochemistry and analysis

Brains from *Tek*-GFP mice were immersion fixed overnight in 4%PFA and then overnight in 30% sucrose before being cut at 50 µm on a freezing microtome. Free floating sections were incubated in pVEGF-R2 antibody (1:300 dilution, Cell Signaling 19A10 Rabbit mAb #2478) in 0.1M PBS for 18 hr, washed and then incubated in Cy5 conjugated secondary antibody (1:400; Invitrogen, A10523) for 4 hr. Confocal image stacks were collected with a 20x objective (NA 0.75) in 2 µm z-steps at a pixel resolution of 0.31 µm/pixel.

To assess vascular pVEGF-R2 signal, all three imaging channels (GFP-labeled endothelium, orange-red DiI/microsphere and far red/Cy5 labeled pVEGF-R2) were split and maximally projected (40 µm). A median filter (radius = 2) was run on each image projection and a threshold was applied (Triangle threshold for GFP and Yen/Moments threshold for DiI/microsphere or pVEGF-2). *Tek*-GFP signal pixels were inverted to create a vascular mask which was then subtracted from the pVEGF-R2 signal to isolate only vascular pVEGF-R2 labeling. For estimating pVEGF-R2 in recanalized (DiI with no microsphere) or obstructed (DiI +microsphere) capillaries, an ROI was drawn 15 µm on either side of a microsphere or center of DiI-labeled capillary and signal pixels were measured. The % coverage of pVEGF-R2 along capillaries was determined by dividing pVEGF-R2 signal pixels by GFP-labeled vascular pixels multiplied by 100. Sham injected mice received DiI-treated saline without microspheres.

## Western blotting

Brains were rapidly removed and the cortex from one hemisphere was dissected and placed immediately in chilled lysis buffer (2 mL/100 mg tissue, CelLytic MT Cell Lysis Reagent for mammalian tissues. Sigma, C322810), and 1 × Halt Proteinase Inhibitor Cocktail and 1 × Halt Phosphatase Inhibitor Cocktail). Samples were sonicated and centrifuged at 14,000 rpm for 15 min at 4°C. Supernatant was then removed and used for gel electrophoresis. The total protein content of the samples was measured with a BCA protein assay kit (Pierce, #23225, 562 nm absorbance). Twenty micrograms of protein were loaded per well and separated on a 8% SDS polyacrylamide gel followed by transfer to PVDF membranes (Bio-Rad Cat# 162–0177) at 40V in transfer buffer (25 mM tris, 192 mM glycine, 20% v/v methanol) overnight at 4°C. Membranes were blocked for 5 hr at room temperature with 5% (w/v) bovine serum albumin (BSA, Sigma, A7906) in tris buffered saline containing Tween 20 (TBST) at room temperature and incubated overnight at 4°C with the following primary antibodies: anti-VEGF-R2(1:1000, Cell Signalling, CS2479s) and anti- β-actin (1:2000) as a loading control (Sigma A-5441) diluted in TBST. Goat anti-rabbit IgG-HRP (Cell Signaling, 7074S) and goat anti-mouse IgG-HRP (KPL, 04-18-15) were used as secondary antibodies. Blots were washed in TBST and incubated with the HRP-conjugated antibody (1:1000) in TBST for 1 hr at room temperature. Blots were developed by enhanced chemiluminescence (BioRad Clarity Western ECL Substrate, Cat# 107–5061) and imaged with a G:BOX Chemi-XR5 (Syngene) gel doc system. Western blot images were processed and quantified by densitometric analysis using Genesys software (version 1.5.3.0, Syngene) and Image Studio Lite (version 5.2, LI-COR Biosciences). Levels of VEGF-R2 were first normalized to the levels of the β-actin loading control and then calculated as fold change of vehicle-injected mice.

## Statistics

Statistical analysis of the data was conducted in GraphPad Prism 5. Relevant two tailed independent or paired t-tests were used to follow up significant one-way or two-way ANOVAs. In some cases, a

priori t-tests were used to compare experimental groups. A repeated measure ANOVA was used for blood flow analysis. Outliers were detected using GraphPad Prism Grubbs test, with an alpha value of 0.05. Two capillaries (from different mice) were identified as outliers and were excluded from analysis in *Figure 3*. Sample sizes for each experiment were based on comparable n values used for similar experiments in the literature. All n's were based on biological replicates. Data are presented as mean ±standard error of the mean (SEM) unless otherwise stated.

## Acknowledgements

The authors would like to thank Dr Kerry Delaney, Natalie Pollock, Emily White and Jesse Spooner for their advice and assistance with experiments, and Drs. Jane Rossant and Andras Nagy for the floxed VEGF-R2 mice. We thank Amanda McLaughlin, Alex Hoggarth, Patrick C Nahirney and Geoff deRosenroll for helpful discussions. This work was supported by operating, salary and equipment grants to CE.B from CIHR, Heart and Stroke Foundation of BC and Yukon, MSFHR, NSERC and CFI.

## Additional information

### Funding

| Funder | Author |
| --- | --- |
| Canadian Institutes of Health Research | Craig E. Brown |
| Natural Sciences and Engineering Research Council of Canada | Craig E. Brown |
| Heart and Stroke Foundation of Canada | Craig E. Brown |

The funders had no role in study design, data collection and interpretation, or the decision to submit the work for publication.

### Author contributions

Patrick Reeson, Conceptualization, Data curation, Software, Formal analysis, Investigation, Methodology, Writing—original draft, Writing—review and editing; Kevin Choi, Formal analysis, Investigation; Craig E Brown, Conceptualization, Supervision, Funding acquisition, Investigation, Writing—original draft, Project administration, Writing—review and editing

### Author ORCIDs

Craig E Brown (iD) https://orcid.org/0000-0002-0076-2414

### Ethics

Animal experimentation: All experiments were conducted according to the guidelines set by the Canadian Council of Animal Care, ARRIVE, and approved by the University of Victoria Animal Care Committee. Protocol 2016-016(2)

### Decision letter and Author response

Decision letter https://doi.org/10.7554/eLife.33670.028
Author response https://doi.org/10.7554/eLife.33670.029

## Additional files

### Supplementary files

• Source code 1. Macro to estimate number of capillaries in 2 P imaging stacks. Code can be executed in either Fiji or ImageJ. It is run on imaging groups of stacks of vasculature in the horizontal

plane (x,y,z, stacks). Output is file with % of vascular pixels for each sub stack as well as number of vascular length pixels.

DOI: https://doi.org/10.7554/eLife.33670.021

• Source code 2. Modelling of capillary loss over time based on experimental data. Code is run in Matlab. Program opens an excel sheet (Data *Reeson, 2018*) with branch order distribution and assigned risk of obstruction (see methods, *Figure 4* and *Figure 4—figure supplement 1*). Output is capillary numbers (by branch order), absolute and relative risk of obstruction and fractional distribution of branch orders, over time.

DOI: https://doi.org/10.7554/eLife.33670.022

• Supplementary file 1. Mod_data excel file, loads into Source Code two to model obstruction risk across branch orders based on the experimentally determined distribution of spontaneous obstructed capillaries.

DOI: https://doi.org/10.7554/eLife.33670.023

• Supplementary file 2. PDF instructions relating to Source code 2.

DOI: https://doi.org/10.7554/eLife.33670.024

• Transparent reporting form

DOI: https://doi.org/10.7554/eLife.33670.025

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
