## [Decision Letter]

Thank you for submitting your article "Mapping the fates of obstructed cerebral capillaries reveals a mechanism for age related capillary loss" for consideration by *eLife*. Your article has been reviewed by three very strong peer reviewers, and the evaluation has been overseen by David Kleinfeld as the Reviewing Editor and David Van Essen as the Senior Editor. The following outside reviewers involved in review of your submission have agreed to reveal their identity: Jaime Grutzendler (Reviewer #1); David A Boas (Reviewer #3). The reviewers have discussed the reviews with one another and the Reviewing Editor has drafted this decision to help you prepare a revised submission.

Summary: Craig Brown and colleagues show that blocking flow in cortical microvessels can lead to substantial pruning of these vessels, exacerbating the initial damage and potentially increasing the risks of hypoxia after the accumulation of occlusions with aging.

Decision: All reviewers, and we as editors, congratulate you on completing an important and thorough study. However, there are many queries raised by the reviewers that require changes to the text before final decision can be made. No additional experiments are needed.

One reoccurring query is on the use of 4 μm microspheres for the block as opposed to large spheres or natural plaques. To paraphrase reviewer 2: "It is essential that the authors clarify that their rates of recanalization, pruning and angiophagy are based on the use of 4um beads and that these analyses would be different based on beads of larger sizes/different composition. Critically, the quantification would almost certainly be different if they were using more natural embolic materials (blood clots or cholesterol emboli), which could markedly increase the retention within capillaries and therefore increase the relative frequency of other recanalization mechanisms different from hemodynamic washout."

There are also qualms about priority. To paraphrase reviewer 2: "The assertion that the authors show angiophagy in capillaries for the first time is incorrect as it has already been shown in Lam et al., (Figure 1D) that capillaries are capable of angiophagy." Likewise, the comment that "With small vessel (capillary) obstructions, angiophagy based recanalization is likely avoided since the capillary can extrude the obstruction back into the circulation which is not possible with large obstructions" is incorrect. As shown in Grutzendler at al., 2014 Figure 1A, large obstructions can also be washout out within hours."

Finally, while the reviewers disagree about the likely contribution of pericytes to the loss of capillaries, we note that pericytes are important in vascular maintenance and integrity and the possible role of pericytes in regards to the failure to recanulate and the onset of subsequent pruning should be discussed.

*Reviewer #1:*

In this manuscript, Reeson et al. use 2-photon in vivo imaging to characterize the incidence and consequences of cortical capillary obstructions. The authors perform time-lapse to follow capillaries blocked spontaneously (>20 min) or with 4µm microspheres injected through the tail vein to track the fate of these vessels. They also model capillary loss with aging and examine the role of VEGF-R2 signaling on rates of recanalization. Overall, the project is well executed and the tracking of capillary fates over time is a solid contribution to the field. There are clear aspects of this study that are interesting, specifically those related to the finding of relatively high rates of microvascular pruning even in vessels that had previously been occluded and underwent recanalization. Also, the results with VEGFR manipulation showing an effect on the pruning outcome are interesting.

Despite this, the authors make conclusions that are not supported by the data. They fail to explain all the caveats associated with using 4µm microbeads and make claims of novelty when previous papers already exist showing similar findings. I strongly suggest that the authors reword the manuscript to better reflect the most interesting and novel aspects i.e. vascular pruning after transient occlusion and the potential role for VEGF rather than the currently vague title. It is also important to reduce emphasis given in the text to the quantification and outcomes of capillary recanalization, as the model used in these experiments is associated with incredibly high washout rates and cannot be used as a gold standard model for capillary obstructions.

1) The authors interpret their 4um microsphere model of capillary obstructions as a robust model to measure capillary recanalization frequency. Use of a 4um microsphere as the sole method to model capillary obstruction is problematic, as the bead is so small that hemodynamic force will almost certainly cause the majority of microspheres to be washed out and does not reflect what happens with more natural embolic composition and slightly larger sizes. Use of slightly larger microspheres (5/6µm) would still result in capillary obstruction, however their rates of washout and transvascular embolic extravasation ("Angiophagy") would likely be completely different from the quantification presented here. It is essential that the authors clarify throughout the manuscript that their rates of recanalization, pruning and angiophagy are based on the use of 4um beads and that these analyses would be different based on beads of larger sizes/different composition. Critically, the quantification would almost certainly be different if they were using more natural embolic materials (blood clots or cholesterol emboli) that are not plastic and smooth as the microspheres, which would markedly increase the retention within capillaries and therefore increase the relative frequency of other recanalization mechanisms different from hemodynamic washout. Thus, as it is, I don't think the relative contributions of each recanalization mechanism can be generalized from the existing data. Authors should add a section discussing the caveats and limitations of the study i.e. the problems associated with using 4um, plastic microspheres that are highly susceptible to washout. Statements such as "Tracking natural or microsphere induced obstructions revealed that most capillaries recanalize by pushing obstructions back into the circulation rather than through the capillary wall" should be removed, as the authors cannot make these conclusions based on the use of 4um plastic spherical beads.

3) The assertion that the authors show angiophagy in capillaries for the first time is incorrect as it has already been shown in Lam et al., 2010 (Figure 1D) that capillaries are capable of angiophagy. This wording should be corrected. Likewise, the comment that "With small vessel (capillary) obstructions, angiophagy based recanalization is likely avoided since the capillary can extrude the obstruction back into the circulation which is not possible with large obstructions" is incorrect. As shown in Grutzendler at al., 2014 Figure 1A, large obstructions can also be washout out within hours. These inaccuracies should be corrected throughout the manuscript.

4) The title of the paper does not reflect the actual novel findings about vascular pruning and VEGF signaling but places too much emphasis on aging and mapping of recanalized capillaries. Title should be reworded to highlight the novel results.

5) Measurements of capillary occlusion and relative rates of microvascular spontaneous stasis have previously been measured by in vivo dynamic imaging (Erdener at al., 2017). This paper should be acknowledged and referenced in the manuscript.

6) The concept of vascular pruning as a result of obstruction/recanalization is interesting and merits further examination. However, the images in Figure 2F have been cropped asymmetrically and it is difficult to see whether or not these are the same vessels over 21 days. Better quality and more examples of pruning images are needed, even just for supplementary data.

7) In Figure 5, the concept of very high and very low VEGF-R2 co-localization with endothelium in recanalized vessels is confusing. Is this due to the duration of how long the obstruction had been there? What causes these differences in expression in recanalized vessels?

8) In Figure 5 did the authors confirm microsphere density/load was similar across groups (saline vs. VEGF-A; vehicle vs. Tamoxifen; vehicle vs. SU5416) acutely e.g. 30 min post microsphere injection? Microsphere load could be heterogeneous (despite injecting the same number of microspheres) and differences in number may have existed at earlier stages which therefore explains the differences in microsphere load seen at 24h, 4d and 21d respectively.

9) More details about density normalization (Figure 5) is needed in the Materials and methods. Exact numbers should be included in the legend or Results section regarding raw numbers of microsphere density. Raw numbers should also be included in Results section (subsection “Superficial and lower order cortical capillaries are prone to obstruction”) regarding number of spontaneous obstructions seen at start of imaging period (persisting >20min) in 10 out of 16 mice.

10) In mice treated with VEGF-A, was there an increase in pruning over time i.e. what were capillary fates in VEGF-A treated mice? Furthermore, was there an increase in disruption in RBC velocity and flux?

*Reviewer #2:*

The study by Reeson and colleagues tracks spontaneous and microsphere-induced obstructions of cortical capillaries that occur preferentially in superficial layers and lower order vessel branches, with a higher prevalence in aged mice. They observed that about 30% of obstructed capillaries undergo pruning, with no evidence of compensatory angiogenesis. Endothelial VEGFR2 was identified as a crucial factor mediating capillary recanalization and pruning. The manuscript's topic is novel and timely, as increasing literature is recognizing the importance of capillary regulation of CBF flow that contributes to neuronal brain health and has relevance to pathophysiology of various neurological disorders. This study would benefit by addressing the following comments:

1) Have the authors considered the contribution of pericytes to their observed capillary constriction and pruning, given the observations that pericytes contribute to "no reflow" phenomena in ischemic conditions (Nature. 2014 Apr 3;508(7494):55-60, Nat Med. 2009 Sep;15(9):1031-7)? At minimum, the possible role of pericytes in regards to the failure to recanulate, delayed vessel constriction, and pruning should be discussed.

2) Is there any blood-brain barrier leakage from the vessels with occlusions? For example, extravascular fibrin/fibrinogen deposits around transient or permanent occlusion sites?

3) There is an implicit assumption that vessels with fluid in them will be labeled with the dyes injected into the vasculature, regardless of whether there is RBC flow. This caveat should be noted, particularly since much of the data relies on this assumption.

4) In Figure 2F, the vessels connected to the pruned vessel appear to deform significantly, particularly at the 15 day time point. How frequently did this phenomenon occur, and were the distances between the two vessels on either side of the pruned vessel permanently altered? It would be interesting to know if the surrounding vessels were remodeled/repositioned in response to the pruning.

5) For the VEGF-A injection experiments, were vascular diameters altered compared to vehicle injections?

6) In the discussion of VEGF-R2 the authors note that VEGF-R2 can upregulate eNOS in endothelial cells. However, it was recently reported that NO is not a major capillary dilator in mouse cortex (Nat Neurosci. 2016 Dec;19(12):1619-1627).

7) Red blood cell (RBC) streaking/velocity measurements:

A) The authors image RBCs "streaking" through the image (first Results paragraph). How long was each vessel imaged in order to determine RBC flow?

B) The details of RBC measurements are not well described or referenced (if the author's technique was based on published work) in the Materials and methods section. This brings up many questions: How were the line scans acquired and at what scan rate? How long was blood flow imaged in order to determine RBC velocity in the vessel segments? If the velocity was measured "by hand" as the methods seem to describe, the method should be described in more detail, as most labs use freely available automated software (i.e. PLoS One. 2012;7(6):e38590 and J Comput Neurosci. 2010 Aug;29(1-2):5-11). The methods indicate only 3 or 9 (it is unclear; subsection “Analysis of vascular structure and flow”) RBC streaks were used for analysis per vessel. With so few measurements per vessel, how did the authors account for natural variation like heart beat when determining velocity (see e.g. PLoS One. 2012;7(6):e38590)?

8) The purpose of the Figure 5B inset is unclear. The same data is shown in the main panel in a more useful organization. Also, the caption should be explained better to describe the "<20%" and ">20%" notations for recanalized vessels.

9) Is there any difficulty with microspheres aggregating during in vivo experiments?

10) The present manuscript only examines vascular responses in the somatosensory cortex and references the expected number of capillary obstructions in the cortex. There may be regional differences in the brain and certain regions (areas of cortex or other regions) may be more or less susceptible to capillary obstructions and/or pruning. Regional vulnerability has relevance to pathophysiology – this is an important concept that should be added to the Discussion.

11) Please explain or correct the use of a 1 way ANOVA with t-test.

12) Please provide more description for Figure 2H in the caption.

13) The difference in vascular density is not very apparent in Figure 4A.

14) Since much of the data is parsed by vessel, not animal, please indicate how many vessels were imaged per animal (typical or range).

15) In many images, it is hard to see the green of the vessel lumen against the magenta of the vascular dye. Perhaps the contrast can be adjusted to better balance the two signals.

16) For immunohistochemistry analysis, how were 50 μm sections projected from 40 μm thick slices (subsection “Phosphorylated VEGF-R2 immunohistochemistry and analysis”)?

*Reviewer #3:*

This study focuses on in vivo measurements in awake and anesthetized mice to detect and quantify the long-lasting (>20 min) capillary obstructions utilizing two-photon microscopy time lapse imaging. They expand their findings with endothelial labeling in stalled capillaries with help of lipophilic microspheres and they also induce a higher number of obstructions by injecting 4-µm fluorescent microspheres. They show that although majority of these obstructions are resolved over time, around 30% of them lead to pruning of capillaries, which can predict age-related decrease in capillary density. Downregulation VEGF signaling in endothelial cells, on the other hand, promote resolution of capillary obstructions, albeit with yet unidentified mechanisms, as the authors appreciate.

This is a very comprehensive study with an excellent design, proper ways of analysis and a very good discussion on findings. The focus of the study is a very interesting and previously underrecognized phenomenon, which can lead to highly innovative future work in cerebral physiology and various pathological models. I nevertheless had some scientific concerns and these points should be addressed for proper conclusion on the findings.

1) It is claimed that injection of 4µm microspheres causes capillary obstructions by themselves. However it is possible that the obstructions are actually caused by blood cells and the microspheres are simply stuck in those capillaries because of lack of flow. The authors did a great deal of work to show that microsphere injection does not lead to microglial activation, physiological variables or blood chemistry profile. However I feel that the most important data to check is the blood cell counts before and after injection of microspheres. An increase in leukocyte count may be happening after microsphere injection, which can readily increase the fraction of capillary stalls. Moreover, another possibility is that these microspheres are phagocytosed by leukocytes and they are actually tracking leukocytes plugging the capillaries and rather than the suggested mechanism of angiophagy, leukocytes may be migrating through the endothelia with fluorescent particles inside. This is a major distinction to be made about the actual pathophysiology.

2) The authors conclude that the obstructions are mainly in lower branch-order capillaries and rather superficially located. In the methods, they describe that they imaged down to 350 µm below cortical surface in vivo. However in Figure 1B they present data on fraction of obstructed vessels down to 1300 µm. Is it possible to get high quality in vivo imaging that deep with FITC and rhodamine fluorophores? Can the relatively lower fraction of stalled capillaries at deeper regions simply be a detection problem because of lower quality? This should be discussed and also the absolute number of capillaries across different depths should be presented in the same panel. Also, while the authors state in the Discussion that this preference of blockages occurring in the lower order branches is consistent with the Erdner and Boas paper, it actually seems to be inconsistent with Figure 5 of that paper which shows that more obstructions occur in the higher order branches near the veins. What gives rise to this discrepancy? Longer lasting blockages as studied here versus more transient blockages studied in Erdener and Boas?

3) It is concluded that anesthesia did not have any effect on the numbers of capillary obstructions. However, it is mentioned that awake mice were also briefly anesthetized before the imaging session before they were fixed to the imaging setup. Could this initial brief anesthesia have long-lasting effects on obstructions even if the mice are awake during the actual imaging? The recently publication by Erdener and Boas, that they cite, does indicate an effect of anesthesia.

---

## [Author Response]

Decision: All reviewers, and we as editors, congratulate you on completing an important and thorough study. However, there are many queries raised by the reviewers that require changes to the text before final decision can be made. No additional experiments are needed.One reoccurring query is on the use of 4 μm microspheres for the block as opposed to large spheres or natural plaques. To paraphrase reviewer 2: "It is essential that the authors clarify that their rates of recanalization, pruning and angiophagy are based on the use of 4um beads and that these analyses would be different based on beads of larger sizes/different composition. Critically, the quantification would almost certainly be different if they were using more natural embolic materials (blood clots or cholesterol emboli), which could markedly increase the retention within capillaries and therefore increase the relative frequency of other recanalization mechanisms different from hemodynamic washout."

We agree that we must be very clear in our manuscript that the rate of recanalization, pruning and angiophagy can be affected by the specific type of embolus used. Although we found similar rates of recanalization and pruning for obstructions induced with microspheres versus those that occurred spontaneously (see Figure 2D and Results subsection “Fates of obstructed cortical capillaries”, last paragraph: 30.2% vessels were pruned when obstructed with microsphere versus 36.7% with spontaneous occlusion, p = 0.48), we acknowledge and discuss work from Grutzendler showing that rates of washout/angiophagy differ between plastic microspheres and natural embolic materials. We have altered the text to reflect this in the Abstract removing the phrase “capillaries recanalized by pushing obstructions back into circulation rather than through the capillary wall”. As well in the Results we state: “However, we should note that we were unable to measure rates of angiophagy associated with spontaneous, naturally occurring emboli, which likely would have influenced the route of recanalization (washout vs. angiophagy).” Lastly in the Discussion under “microsphere model of obstruction”, we now state: “However, there are important caveats that should be noted with the microsphere model. […] For example, Lam et al. showed different rates of washout versus angiophagy between natural emboli compared to microspheres (Lam et al., 2010).”

There are also qualms about priority. To paraphrase reviewer 2: "The assertion that the authors show angiophagy in capillaries for the first time is incorrect as it has already been shown in Lam et al., 2010 (Figure 1D) that capillaries are capable of angiophagy." Likewise, the comment that "With small vessel (capillary) obstructions, angiophagy based recanalization is likely avoided since the capillary can extrude the obstruction back into the circulation which is not possible with large obstructions" is incorrect. As shown in Grutzendler at al., 2014 Figure 1A, large obstructions can also be washout out within hours."

We agree and appreciate the correction. We have changed the text in the Discussion, stating “Our finding of an angiophagy-like process where endothelial cells engulf and extrude emboli through the vessel wall replicates previous findings of Lam et al. (Lam et al., 2010; Grutzendler et al., 2014). […] For example, larger occlusions (> 4 µm) would have more severe consequences such as hypoxia and cell death (Tennant and Brown, 2013) which can trigger major structural changes in the vascular endothelium through hypoxia inducible factors.” and we have removed the statement “With small vessel (capillary) obstructions, angiophagy based recanalization is likely avoided since the capillary can extrude the obstruction back into the circulation (see Figure 2—video 1), which is not possible with large obstructions.”

Finally, while the reviewers disagree about the likely contribution of pericytes to the loss of capillaries, we note that pericytes are important in vascular maintenance and integrity and the possible role of pericytes in regards to the failure to recanulate and the onset of subsequent pruning should be discussed.

This is an excellent point, and one that we had considered ourselves. However, in order to do this important topic justice, a full length study would be required which we think would be best left for future investigation. We now discuss this topic in the “Mechanisms of recanalization” section of the Discussion, adding “Another possibility that will require further study is whether pericytes facilitate recanalization or pruning. […] Pericytes could also play a significant role in vessel pruning since they are hotspots of matrix metalloprotease activity associated with BBB disruption (Underly et al., 2016), which is needed for remodeling of the extracellular matrix and basement membrane around capillaries.”

Reviewer #1:In this manuscript, Reeson et al. use 2-photon in vivo imaging to characterize the incidence and consequences of cortical capillary obstructions. The authors perform time-lapse to follow capillaries blocked spontaneously (>20 min) or with 4µm microspheres injected through the tail vein to track the fate of these vessels. They also model capillary loss with aging and examine the role of VEGF-R2 signaling on rates of recanalization. Overall, the project is well executed and the tracking of capillary fates over time is a solid contribution to the field. There are clear aspects of this study that are interesting, specifically those related to the finding of relatively high rates of microvascular pruning even in vessels that had previously been occluded and underwent recanalization. Also, the results with VEGFR manipulation showing an effect on the pruning outcome are interesting.Despite this, the authors make conclusions that are not supported by the data. They fail to explain all the caveats associated with using 4µm microbeads and make claims of novelty when previous papers already exist showing similar findings. I strongly suggest that the authors reword the manuscript to better reflect the most interesting and novel aspects i.e. vascular pruning after transient occlusion and the potential role for VEGF rather than the currently vague title. It is also important to reduce emphasis given in the text to the quantification and outcomes of capillary recanalization, as the model used in these experiments is associated with incredibly high washout rates and cannot be used as a gold standard model for capillary obstructions.1) The authors interpret their 4um microsphere model of capillary obstructions as a robust model to measure capillary recanalization frequency. Use of a 4um microsphere as the sole method to model capillary obstruction is problematic, as the bead is so small that hemodynamic force will almost certainly cause the majority of microspheres to be washed out and does not reflect what happens with more natural embolic composition and slightly larger sizes. Use of slightly larger microspheres (5/6µm) would still result in capillary obstruction, however their rates of washout and transvascular embolic extravasation ("Angiophagy") would likely be completely different from the quantification presented here. It is essential that the authors clarify throughout the manuscript that their rates of recanalization, pruning and angiophagy are based on the use of 4um beads and that these analyses would be different based on beads of larger sizes/different composition. Critically, the quantification would almost certainly be different if they were using more natural embolic materials (blood clots or cholesterol emboli) that are not plastic and smooth as the microspheres, which would markedly increase the retention within capillaries and therefore increase the relative frequency of other recanalization mechanisms different from hemodynamic washout. Thus, as it is, I don't think the relative contributions of each recanalization mechanism can be generalized from the existing data. Authors should add a section discussing the caveats and limitations of the study i.e. the problems associated with using 4um, plastic microspheres that are highly susceptible to washout. Statements such as "Tracking natural or microsphere induced obstructions revealed that most capillaries recanalize by pushing obstructions back into the circulation rather than through the capillary wall" should be removed, as the authors cannot make these conclusions based on the use of 4um plastic spherical beads.

Please see our first response to the decision comments above..

3) The assertion that the authors show angiophagy in capillaries for the first time is incorrect as it has already been shown in Lam et al., 2010 (Figure 1D) that capillaries are capable of angiophagy. This wording should be corrected. Likewise, the comment that "With small vessel (capillary) obstructions, angiophagy based recanalization is likely avoided since the capillary can extrude the obstruction back into the circulation which is not possible with large obstructions" is incorrect. As shown in Grutzendler at al., 2014 Figure 1A, large obstructions can also be washout out within hours. These inaccuracies should be corrected throughout the manuscript.

Please see our second response to the decision comments above.

4) The title of the paper does not reflect the actual novel findings about vascular pruning and VEGF signaling but places too much emphasis on aging and mapping of recanalized capillaries. Title should be reworded to highlight the novel results.

We agree, finding a title that is both succinct and encompassing of the most novel aspects is always challenging. We have changed the title to better reflect an emphasis on VEGF signaling.

“VEGF signaling regulates the fate of obstructed capillaries in mouse cortex”

*5) Measurements of capillary occlusion and relative rates of microvascular spontaneous stasis have previously been measured by* in vivo *dynamic imaging (Erdener at al., 2017). This paper should be acknowledged and referenced in the manuscript.*

We are aware of the excellent work by Erdener at al. and cited it in our original submission. We have also added another reference to this paper in the Introduction: “A recent study used Optical Coherence Tomography to estimate that over 9 minutes, up to 7.5% of capillaries experienced a stall (Erdener et al., 2017).”

6) The concept of vascular pruning as a result of obstruction/recanalization is interesting and merits further examination. However, the images in Figure 2F have been cropped asymmetrically and it is difficult to see whether or not these are the same vessels over 21 days. Better quality and more examples of pruning images are needed, even just for supplementary data.

We can see how it may appear to the reviewer that the images in Figure 2F were cropped asymmetrically, however that is due to the occasional distortion of adjacent vessels during pruning (see response to Reviewer 2 comment 4). To provide additional clarity we have added several new examples of obstruction/recanalization in Figure 2—figure supplement 1A, Figure 2—figure supplement 2B.

7) In Figure 5, the concept of very high and very low VEGF-R2 co-localization with endothelium in recanalized vessels is confusing. Is this due to the duration of how long the obstruction had been there? What causes these differences in expression in recanalized vessels?

The wide disparity of activated VEGF-R2 (p-VEGF-R2) colocalization with recanalized vessels 3 hours post-microsphere injection surprised us as well. Specifically, we noted that rather than a gaussian range of values, we instead found two very distinct, non-overlapping, populations with very high or low VEGF-R2 colocalization (Figure 5B insert). While our DiI labelling technique allowed us to spatially “stamp” recanalized vessels, it had limitations in the time domain. Assuming a minimum time required for DiI to label a vessel (<30 minutes), we could not know precisely how long the obstructed vessel had regained flow (likely anywhere from just minutes to 2.5 hours). Based on previous studies showing that VEGF-R2 signaling is sensitive to shear stress (increased flow leads to increased phosphorylated VEGF-R2) and our VEGF-R2 inhibitor/ gene knockdown experiments showing that reducing VEGF-R2 enhances recanalization, we suspect that low pVEGF-R2 colocalization would correlate with vessels that had recently regained blood flow (with respect to the time of killing the mouse). This would also explain why most obstructed vessels have a natural tendency to recanalize, as well as why pVEGF-R2 colocalization was significantly lower in vessels 30 minutes after microsphere injection. With respect to recanalized vessels with high pVEGF-R2 colocalization 3 hours after injection, we speculate these are vessels that had regained flow for an extended period of time, thereby allowing shear stress induced VEGF-R2 phosphorylation to increase substantially.

8) In Figure 5 did the authors confirm microsphere density/load was similar across groups (saline vs. VEGF-A; vehicle vs. Tamoxifen; vehicle vs. SU5416) acutely e.g. 30 min post microsphere injection? Microsphere load could be heterogeneous (despite injecting the same number of microspheres) and differences in number may have existed at earlier stages which therefore explains the differences in microsphere load seen at 24h, 4d and 21d respectively.

This is an important point, and we thank the reviewer for bringing it up. It is an assumption in our histological experiments (where microsphere density is quantified) that even with perfectly identical injections of microspheres with control and experimental groups run in parallel, the starting obstruction load in the brain could differ between experimental groups. With VEGF-R2 inhibitor experiments, mice were randomly assigned to SU5416 or vehicle groups 30 mins after microsphere injection, thus there was no initial bias in assigning treatment groups. However to confirm this, we examined our in vivodata at the first imaging point (+30 min post-injection) and found no difference in initial microsphere density (652 ± 224 microspheres / mm^3^ for vehicle injected mice versus 562 ± 200 microspheres / mm^3^ for SU5416 treated mice, t_(6)_ = 0.25, p = 0.8).

For genetic knockdown of VEGF-R2, Tamoxifen was given prior to injection of microspheres, possibly influencing the number of microspheres reaching the brain. However, our in vivo imaging data 30 min post microsphere injection did not show a significant difference in the number of initial obstructions (509 ± 234 microspheres / mm^3^ for vehicle injected mice versus 328 ± 70 microspheres / mm^3^ for Tamoxifen treated, t_(10)_ = 2.0, p =0.07). We now include these analyses in the last paragraph of the results. More importantly, by following the same obstructed vessels over time in vivo(for both VEGF-R2 inhibition and knockdown experiments),initial microsphere load was not a factor in explaining the outcome of an obstructed vessel. In all experiments, we observed the same strong effects in each experimental group (regardless of variability in initial load) for both histological and in vivo time lapse imaging studies. In summary, we are confident that variability in initial microsphere density could not explain our converging lines of evidence indicating a key role for VEGF-R2 in capillary recanalization.

9) More details about density normalization (Figure 5) is needed in the Materials and methods. Exact numbers should be included in the legend or Results section regarding raw numbers of microsphere density. Raw numbers should also be included in Results section (subsection “Superficial and lower order cortical capillaries are prone to obstruction”) regarding number of spontaneous obstructions seen at start of imaging period (persisting >20min) in 10 out of 16 mice.

We have amended the Materials and methods sections to better clarify the normalization procedure, “To limit variability in total numbers of microspheres injected, mice were run in balanced groups of control and experimental conditions and received injections from the same diluted stock of microspheres. […] Normalized ratios were then averaged across cohorts (typically each experiment consisted of 2-4 cohorts).” and have added raw microsphere density to the Figure 5 legend. With respect to spontaneous obstructions, we have added the raw number to both the legend and in the Results section, “(persisted > 20 min, 20 obstructed capillaries in 10 animals, range of 1-4 per mouse)”.

10) In mice treated with VEGF-A, was there an increase in pruning over time i.e. what were capillary fates in VEGF-A treated mice? Furthermore, was there an increase in disruption in RBC velocity and flux?

This is an intriguing question, as our data shows that 24 hours after microsphere injections, VEGF-A treated mice had a significantly higher number of microspheres in the cortex. However, we do not have a concrete answer to this question since we did not undertake these challenging experiments of following individual obstructions over a 21 day period. This is largely due to the challenging task of chronic i.c.v. infusion of VEGF (with cannula hardware cemented to the skull) in mice with hardware needed for long-term imaging through a cranial window (metallic ring, dental cement etc.). However, with that said, we do know that there is a high probability for long-lasting obstructions (in this case lasting at least 24 hours) to lead to vessel pruning.

Reviewer #2:The study by Reeson and colleagues tracks spontaneous and microsphere-induced obstructions of cortical capillaries that occur preferentially in superficial layers and lower order vessel branches, with a higher prevalence in aged mice. They observed that about 30% of obstructed capillaries undergo pruning, with no evidence of compensatory angiogenesis. Endothelial VEGFR2 was identified as a crucial factor mediating capillary recanalization and pruning. The manuscript's topic is novel and timely, as increasing literature is recognizing the importance of capillary regulation of CBF flow that contributes to neuronal brain health and has relevance to pathophysiology of various neurological disorders. This study would benefit by addressing the following comments:1) Have the authors considered the contribution of pericytes to their observed capillary constriction and pruning, given the observations that pericytes contribute to "no reflow" phenomena in ischemic conditions (Nature. 2014 Apr 3;508(7494):55-60, Nat Med. 2009 Sep;15(9):1031-7)? At minimum, the possible role of pericytes in regards to the failure to recanulate, delayed vessel constriction, and pruning should be discussed.

This is an excellent point, and one that we had considered ourselves. However, in order to do this important topic justice, a full length study would be required which we think would be best left for future investigation. We now discuss this topic in the “Mechanisms of recanalization” section of the Discussion, adding “Another possibility that will require further study is whether pericytes facilitate recanalization or pruning. […] Pericytes could also play a significant role in vessel pruning since they are hotspots of matrix metalloprotease activity which is needed for remodeling of the extracellular matrix and basement membrane around capillaries (Forde et al., 2002).”

2) Is there any blood-brain barrier leakage from the vessels with occlusions? For example, extravascular fibrin/fibrinogen deposits around transient or permanent occlusion sites?

We occasionally noticed labelled plasma leak from obstructed capillaries at or near the site of an obstruction, appearing as a small extravascular plume 30 min – 1.5 hour after the initial obstruction. However, these were relatively rare occurrences for both microsphere (5 vessels out of 162 total) and spontaneously obstructed capillaries (2 vessels out of 21 total). For VEGFR-2 inhibition, only 1 out of 93 vessels showed BBB disruption and none were detected in VEGF-R2 knockdown experiments. We did not find any relationship between BBB disruption and whether an obstruction was cleared or pruned.

3) There is an implicit assumption that vessels with fluid in them will be labeled with the dyes injected into the vasculature, regardless of whether there is RBC flow. This caveat should be noted, particularly since much of the data relies on this assumption.

This is an important caveat that should be explicitly stated in the text. We also noted that while using our *Tek*-GFP labelled endothelial mice, we never saw completely empty lumens (i.e. *Tek*-GFP labelled vessels with no dye infused plasma). Even in the cases of obstruction and pruning (with no RBC flow) some labelled plasma always diffused into parts of the obstructed vessel. We have added a sentence in the Microsphere model section of the Discussion: “Our measurements were based on intravenous injection of fluorescent dyes, therefore labelling flowing capillaries in the brain with very high signal to noise. However, this could slightly overestimate the density of patent capillaries since injected plasma dyes also filled stalled capillaries given that we never found a *Tek*-GFP labelled capillary without fluorescent dye present in it.”

4) In Figure 2F, the vessels connected to the pruned vessel appear to deform significantly, particularly at the 15 day time point. How frequently did this phenomenon occur, and were the distances between the two vessels on either side of the pruned vessel permanently altered? It would be interesting to know if the surrounding vessels were remodeled/repositioned in response to the pruning.

This is an interesting observation. We agree that in the act of pruning, adjacent vessels could appear somewhat deformed, such as pulling together. Therefore, we measured the shortest straight distance (in 3D) between starting branch points and final distances (before and after the adjoining capillary was pruned). The distance between 33 pairs of vessels in 10 mice was measured where the same points (after pruning) could be unambiguously determined by landmarks and endothelial cell bodies (range 1-7 pairs of vessels per mouse). We found ~36% (12/33) were slightly further apart after pruning (average change in distance from time 0 to +21 days: +5.77 ± 8.5 μm, range 0.4-29 μm). Conversely ~64% of vessels were closer together after pruning (21/33, average change in distance from time 0 to +21 days: -8.67 ± 7.7 μm, range 1.3-32 μm). Statistically speaking, there was no significant difference in the average distance between vessel segments before or 21 days after pruning. This data is now included in the Results section and Figure 2—figure supplement 1: “During pruning we occasionally found that the adjacent capillary segments appeared closer together or further apart. However, by measuring inter-capillary distance at time 0 and +21 days, on average we found no significant lasting deformation of the newly separated capillaries compared to controls (Figure 2—figure supplement 1).”

5) For the VEGF-A injection experiments, were vascular diameters altered compared to vehicle injections?

Unfortunately, we were unable to answer this question with our VEGF injection experiments. We were primarily interested in the effects of VEGF on microsphere clearance and used a histology approach. If we were to assess this question 24 hours after VEGF injection in post mortem fixed tissue, any VEGF induced changes to vessel diameter would likely have dissipated and/or be obscured by the loss of blood pressure and shrinkage related to post-mortem tissue processing. While this is a question we are still interested in answering, particularly as there is evidence that VEGF can alter blood flow (Hein, T. W., et al. (2015). Investigative Ophthalmology and Visual Science 56(9): 5381-5389), properly answering this question would require technically challenging experiments combining in vivo imaging with i.c.v. cannula infusion of VEGF. However, we did note in the manuscript that SU5416 based inhibition of VEGF-R2 signaling did not have a detectable effect on capillary width in our in vivo imaging experiments. Specifically, there was no significant difference in average capillary width when measured before and 90 mins after SU5416 injection (paired t-test t_(29)_=0.9, p= 0.37). We have added this data to Figure 5—figure supplement 2G.

6) In the discussion of VEGF-R2 the authors note that VEGF-R2 can upregulate eNOS in endothelial cells. However, it was recently reported that NO is not a major capillary dilator in mouse cortex (Nat Neurosci. 2016 Dec;19(12):1619-1627).

We have added this caveat to the text in the Mechanisms of Recanalization section of the Discussion “However a recent study has questioned the role of NO as a major capillary dilator (Mishra et al., 2016), thus other pathways may be involved”

7) Red blood cell (RBC) streaking/velocity measurements:A) The authors image RBCs "streaking" through the image (first Results paragraph). How long was each vessel imaged in order to determine RBC flow?

Each vessel was imaged for at least ~6.5 seconds (3.2475 seconds/frame, 2um steps) during initial x,y,z imaging. Any vessel that had no observable streaking within this imaging volume was then followed up by linescans (~3 sets of line scans 30 seconds apart) to confirm the lack of RBC flow. Then each region was again imaged 20 min later in x,y,z stack (again ~6.5 seconds minimum for a ~4 μm vessel perfectly in xy plane) and then again followed up by linescans to be consider obstructed for greater than 20 min.

B) The details of RBC measurements are not well described or referenced (if the author's technique was based on published work) in the Materials and methods section. This brings up many questions: How were the line scans acquired and at what scan rate? How long was blood flow imaged in order to determine RBC velocity in the vessel segments? If the velocity was measured "by hand" as the methods seem to describe, the method should be described in more detail, as most labs use freely available automated software (i.e. PLoS One. 2012;7(6):e38590 and J Comput Neurosci. 2010 Aug;29(1-2):5-11). The methods indicate only 3 or 9 (it is unclear; subsection “Analysis of vascular structure and flow”) RBC streaks were used for analysis per vessel. With so few measurements per vessel, how did the authors account for natural variation like heart beat when determining velocity (see e.g. PLoS One. 2012;7(6):e38590)?

We have clarified our linescan analysis in the Materials and methods stating: “To measure RBC velocity, two blinded researchers chose 3 equally spaced apart RBC streaks from each line scan (each linescan was on average 30.83 ± 1 ms per scan, 2 μm/pixel) and using built in functions of Olympus Fluoview software measured the inverse slope (Δ time/ Δ distance). […] Therefore, each single vessel RBC velocity at each timepoint was the average of 18 independent blinded measurements (2 experimenters’ x 3 linescans x 3 RBC slopes).”

To further verify that our measurements were comparable to those achieved by automated programs, we randomly selected 20 linescans to measure using the available MATLAB program from PLoS One. 2012;7(6):e38590. From this set we compared average manual versus automated RBC velocity and found no difference (see Author response image 1). Two linescans were excluded due to poor signal to noise that was incompatible with the automated program and gave physiologically impossible values. While the short duration of each linescan (average 30.83 ± 1 ms) prevents us from excluding the effects of cardiac activity, we did not find any treatment related changes in heart rate (Figure 5—figure supplement 1-2). Thus, any effects of heartbeat would be equally dispersed across experimental groups.

8) The purpose of the Figure 5B inset is unclear. The same data is shown in the main panel in a more useful organization. Also, the caption should be explained better to describe the "<20%" and ">20%" notations for recanalized vessels.

We apologize for the lack of clarity regarding Figure 5. As mentioned in our response to reviewer 1 (comment 7), the purpose of the insert in 5B was to show that recanalized capillaries appeared to form to distinct populations, those with very high or low pVEGF-R2 colocalization. These very different distributions were used to justify splitting recanalized vessels into 2 populations (those < 20% vs those >20%). We have added clarification to the figure legend “Inset shows the distribution of pVEGF-R2 colocalization values for all recanalized capillaries. Based on this distribution we used a 20% cut off (dotted line) to separate the 2 distinct populations. “.

*9) Is there any difficulty with microspheres aggregating during* in vivo *experiments?*

In rare cases two microspheres would be found obstructing the same vessel, however these vessels were always excluded from analysis. We have added this point in the Materials and methods.

10) The present manuscript only examines vascular responses in the somatosensory cortex and references the expected number of capillary obstructions in the cortex. There may be regional differences in the brain and certain regions (areas of cortex or other regions) may be more or less susceptible to capillary obstructions and/or pruning. Regional vulnerability has relevance to pathophysiology – this is an important concept that should be added to the Discussion.

This is an excellent point. Although we have shown in Figure 1—figure supplement 4 relatively even distribution in dorsal cortex (from -2mm to +1.5mm relative to bregma), this is admittedly a simplified axis that doesn’t take into account possible variability within dorsal cortex or regional differences that may occur in more anterior, posterior and lateral cortical regions. Given this possibility and its relevance to pathology, we have added this point to the Discussion under Microsphere model of obstruction section: “While we did not find any differences in the anterior to posterior distribution of microspheres in dorsal cortex (from -2mm to +1.5mm relative to bregma), this doesn’t take into account regional differences that may exist in more anterior, posterior and lateral cortical regions. Since regional vulnerability to capillary obstruction has relevance to pathophysiology, future work will be needed to characterize the risk of obstruction and pruning across the cortical mantle or subcortical regions.”

11) Please explain or correct the use of a 1 way ANOVA with t-test.

In all cases we first performed a 1-way ANOVA using Graphpad Prism software, only significant 1-way ANOVAs where followed up by relevant independent 2 tailed t-test. We have clarified this in the Materials and methods section.

12) Please provide more description for Figure 2H in the caption.

We have done this adding to the caption “H) Summary of microsphere obstructed capillary fates 21 days after injection (n=14 mice, 162 obstructions). Numbers in parentheses indicate how many capillaries out of 162 total capillaries underwent pruning, angiophagy or recanalized by extrusion. Average 11 capillaries per mouse, range 6-26.”

13) The difference in vascular density is not very apparent in Figure 4A.

The reviewer is correct in noting that difference in vascular density is not stark in Figure 4A. However, since the he overall loss of capillaries was ~ 10%, an obvious difference would be difficult to detect by eye in any representative images.

14) Since much of the data is parsed by vessel, not animal, please indicate how many vessels were imaged per animal (typical or range).

We have added this information (average number of capillaries and range per mouse) to the relevant figure legends.

15) In many images, it is hard to see the green of the vessel lumen against the magenta of the vascular dye. Perhaps the contrast can be adjusted to better balance the two signals.

We have adjusted the contrast in images to better balance the two signals.

16) For immunohistochemistry analysis, how were 50 μm sections projected from 40 μm thick slices (subsection “Phosphorylated VEGF-R2 immunohistochemistry and analysis”)?

Thank for pointing out this error, we have changed it to the correct thickness of 50 μm and a projection of 40 μm.

Reviewer #3:This study focuses on in vivo measurements in awake and anesthetized mice to detect and quantify the long-lasting (>20 min) capillary obstructions utilizing two-photon microscopy time lapse imaging. They expand their findings with endothelial labeling in stalled capillaries with help of lipophilic microspheres and they also induce a higher number of obstructions by injecting 4-µm fluorescent microspheres. They show that although majority of these obstructions are resolved over time, around 30% of them lead to pruning of capillaries, which can predict age-related decrease in capillary density. Downregulation VEGF signaling in endothelial cells, on the other hand, promote resolution of capillary obstructions, albeit with yet unidentified mechanisms, as the authors appreciate.This is a very comprehensive study with an excellent design, proper ways of analysis and a very good discussion on findings. The focus of the study is a very interesting and previously underrecognized phenomenon, which can lead to highly innovative future work in cerebral physiology and various pathological models. I nevertheless had some scientific concerns and these points should be addressed for proper conclusion on the findings.1) It is claimed that injection of 4µm microspheres causes capillary obstructions by themselves. However it is possible that the obstructions are actually caused by blood cells and the microspheres are simply stuck in those capillaries because of lack of flow. The authors did a great deal of work to show that microsphere injection does not lead to microglial activation, physiological variables or blood chemistry profile. However I feel that the most important data to check is the blood cell counts before and after injection of microspheres. An increase in leukocyte count may be happening after microsphere injection, which can readily increase the fraction of capillary stalls. Moreover, another possibility is that these microspheres are phagocytosed by leukocytes and they are actually tracking leukocytes plugging the capillaries and rather than the suggested mechanism of angiophagy, leukocytes may be migrating through the endothelia with fluorescent particles inside. This is a major distinction to be made about the actual pathophysiology.

Regarding blood cell counts, we did consider this as an important variable. Our blood analysis (shown in Figure 1—figure supplement 2J) measured in microsphere injected mice (at 4 and 21 days post-injection) did not reveal any significant difference in mean hematocrit levels, suggesting a large-scale change in circulating blood cells did not occur. However, we certainly agree that we cannot rule out the possibility that leukocytes engulf and move microspheres across the endothelium and now state this in the Discussion section on Mechanisms of Recanalization: “Our finding of an angiophagy-like process where endothelial cells engulf and extrude emboli through the vessel wall replicates previous findings of Grutzendler lab (Lam et al., 2010; Grutzendler et al., 2014). […] Due to the low incidence (2%) of angiophagy-like events in our microsphere model, we could not extensively characterize this process.”

2) The authors conclude that the obstructions are mainly in lower branch-order capillaries and rather superficially located. In the methods, they describe that they imaged down to 350 µm below cortical surface in vivo. However in Figure 1B they present data on fraction of obstructed vessels down to 1300 µm. Is it possible to get high quality in vivo imaging that deep with FITC and rhodamine fluorophores? Can the relatively lower fraction of stalled capillaries at deeper regions simply be a detection problem because of lower quality? This should be discussed and also the absolute number of capillaries across different depths should be presented in the same panel. Also, while the authors state in the Discussion that this preference of blockages occurring in the lower order branches is consistent with the Erdner and Boas paper, it actually seems to be inconsistent with Figure 5 of that paper which shows that more obstructions occur in the higher order branches near the veins. What gives rise to this discrepancy? Longer lasting blockages as studied here versus more transient blockages studied in Erdener and Boas?

We apologize for the lack of clarity in Figure 1. Data in Figure 1B was derived from post mortem histology, thus the full cortex was accessible in coronal slices. We have clarified this in the Figure 1 legend. Furthermore, we have added absolute and relative capillary density values for each depth: “B) Distribution of microsphere obstructed capillaries as it relates to depth from the pial surface determined by confocal imaging from post-mortem brain sections. […] Black line indicates fraction of capillaries by depth (error bars are 95% CI) as well as raw numbers of capillaries / mm^3^ by depth are provided in parentheses.” We should note that these relative capillary density fractions closely align with previous estimates from Kleinfeld lab (Tsai et al., 2009 and Blinder et al., 2013).

As to the discrepancy with Erdener and Boas, we meant to suggest that there are similarities in the distribution of obstructions at the arteriole end where the incidence of obstruction peaks at the third order branch with a gaussian distribution around this peak. As for the distribution on the venous side (shown in Erdener and Boas in Figure 5), our data analysis did not address this aspect since we only quantified obstructions as a function of branch order from the arteriole side. We now clarify in the figure legend that we quantified arteriole branch order and mention the discrepancies in the first paragraph of the Discussion: “However, the risk of obstruction was not equally distributed as superficial capillaries of lower branch orders were at highest risk (Figure 1C). While this distribution of obstructed capillaries (as a function of arteriole branch order) bears similarity to a recent study (Erdener et al., 2017), we did not quantify risk relative to venous branch order which may in fact have a higher risk for obstruction (Erdener et al., 2017).”

3) It is concluded that anesthesia did not have any effect on the numbers of capillary obstructions. However, it is mentioned that awake mice were also briefly anesthetized before the imaging session before they were fixed to the imaging setup. Could this initial brief anesthesia have long-lasting effects on obstructions even if the mice are awake during the actual imaging? The recently publication by Erdener and Boas, that they cite, does indicate an effect of anesthesia.

Since our animal ethics protocol requires brief anesthesia to intravenously inject microspheres, we could not systematically determine whether brief exposure to isoflurane to inject beads (~5-10min) would have altered the number of obstructions relative to those without any anesthesia. Therefore, we acknowledge the possibility that anesthesia could have impacted the number of obstructions and cited Erdener at al. and another recent study (Santisakultarm et al., 2016). This important issue is touched on in the microsphere model of obstruction section in the Discussion: “A second caveat that could not be avoided was that we were unable to inject microspheres without the use of anesthesia. […] Therefore we cannot completely rule out a possible effect of even brief isoflurane exposure (in awake imaging or histology experiments) on microsphere obstruction density, as others have reported anesthetic effects on capillary stalling ((Erdener et al., 2017; Santisakultarm et al., 2016).”